# Little evidence for a role of facial mimicry in the transmission of stress from parents to adolescent children
Jost Ulrich Blasberg [1] ✉, Philipp Kanske [2] & Veronika Engert [1,3,4]

Empathic stress, the spontaneous reproduction of psychosocial stress by mere observation, has been shown to occur between strangers, romantic partners and in mother-child dyads. However, the mechanisms by which stress is transmitted have yet to be understood. We investigated whether facial mimicry modulates the transmission of psychosocial stress. Adolescents (13-16 years old) observed their mothers or fathers (N = 77) undergo a standardized laboratory stressor. Parents' and adolescents' faces were videotaped during the stress task and dyads simultaneously provided multiple samples of subjective stress, heart rate, heart rate variability (HRV), and salivary cortisol. The degree to which adolescents mimicked their parents' facial expressions was calculated in a multi-step procedure based on windowed-cross-lagged-regressions. To integrate the correlational structure of mimicry across different facial action units (AU), an exploratory factor analysis was employed. The solution revealed a two-factor model, constructed of a positive latent factor subsuming mimicked action units associated with the act of smiling and a negative latent factor, subsuming mimicked action units used for various negative emotions. None of the stress markers were significantly associated with the extracted latent factors indexing mimicry between parents and adolescents, providing no statistically significant evidence for an association between facial mimicry and stress-transmission in the parent-adolescent dyad. Bayes Factors generally indicated moderate evidence for a lack of association with the positive and anecdotal evidence for a lack of association with negative latent mimicry factors. In conclusion, our approach to video-based mimicry calculation showed promising results in that mimicry of positive and negative emotions could be detected, albeit no evidence for a link to actual empathic stress transmission in the laboratory was found.

Empathy, the ability to reproduce others' affective states, is a highly adaptive mechanism involved in sharing information and motivating appropriate behavior[1]. Beyond the sharing of subjective experience, empathy entails the reproduction of others' physiological and neural states. This can occur through affective empathy, which is the mere reproduction of another's feeling, or cognitive empathy, which combines affect sharing with mentalizing about another's thoughts and feelings[2]. Empathy also extends to psychosocial stress[3,4], defined as the perturbation of allostasis due to challenges characterized by socio-evaluative threat and uncontrollability[5]. Accordingly, *empathic stress* refers to the reproduction of an acute stress response in the passive observer of a directly stressed target individual, involving autonomic nervous system (ANS), hypothalamic-pituitary-adrenal (HPA) axis, and subjective-psychological activation. Several studies have provided evidence for the existence of empathic stress on different levels of the stress system and in distinct participant samples (for reviews, see refs. 3,6,7). Aiming to understand the possible mediating role of facial mimicry in empathic stress occurrence, we here investigate the degree to which adolescents mimic their acutely stressed parents' facial muscle behavior.

Buchanan and colleagues[8] first found evidence for *stress-resonance*, the proportional release of cortisol with the directly stressed targets, in laboratory assistants administering a psychosocial laboratory stressor. These

[1]Institute of Psychosocial Medicine, Psychotherapy and Psychooncology, Jena University Hospital, Friedrich-Schiller University, Jena, Germany. [2]Clinical Psychology and Behavioral Neuroscience, Faculty of Psychology, Technische Universität Dresden, Dresden, Germany. [3]German Center for Mental Health (DZPG), partner site Halle-Jena-Magdeburg, Jena, Germany. [4]Center for Intervention and Research in adaptive and maladaptive brain Circuits underlying mental health (C-I-R-C), Halle-Jena-Magdeburg, Jena, Germany. ✉e-mail: Jost.Blasberg@med.uni-jena.de

findings were subsequently broadened to observers not actively engaged in the stress task[9], showing a higher probability of cortisol stress-resonance when targets and observers were in a romantic relationship (vs. strangers), and when observers watched through a one-way mirror (vs. via virtual camera feed).

Due to its high relevance for the development of the human stress system across childhood and puberty[10], the mother–child bond is of particular interest in the study of stress-resonance[11–13]. In this context, the pubertal stress recalibration hypothesis[14] states that adolescence serves as a sensitive period, providing both a chance to rejuvenate or worsen the effects of early life adversity. Thus, in an ideal familial scenario, empathic stress provides the necessary energy to motivate appropriate helping behavior and jointly alleviate a stressor at hand[3], providing opportunities to safely experience adversity together. If, however, existing stressors cannot be alleviated and parental chronic stress is shared across longer periods of time, the stress in the family system might rather spread than dissipate. This would put children at risk of hazardous living circumstances and the early development of stress-associated disease[10].

Illustrating the occurrence of stress-resonance already at an early developmental stage, Waters and colleagues[11] found mother–infant heart rate synchronization in acutely stressed mothers holding their children. We corroborated these findings in 8–12-year-old children[13], providing evidence for empathic stress occurrence across middle-childhood. In detail, mothers either completed a standardized laboratory stressor, the Trier social stress test (TSST)[15], or a stress-free control task while their children were watching. Stressed mother–child dyads showed both stress-resonance and *vicarious stress reactivity*, defined as an observer stress response irrespective of target stress activation[9]. Thus, stress group children exhibited elevated levels of subjective stress experience, a higher proportion of physiologically significant cortisol responders (>0.5 nmol/l[16]), and a greater decrease in high-frequencey heart rate variability (HF-HRV) depending on higher levels of trait cognitive empathy. Stressed children's HF-HRV also declined proportionally to that of their mothers, suggesting stress-resonance in terms of parasympathetic nervous system activity[13].

Although the phenomenon of empathic stress has been successfully replicated in multiple studies, we know very little about the intricacies of *how* stress is shared on a behavioral level. Mimicry, the spontaneous reproduction of changes in facial expressions, has been suggested as a subliminal precursor of emotional contagion and empathy[17–20], and may therefore play a role in the transmission of stress. The emotional mimicry in context model[21] theorizes that rather than constituting context-free expression reproduction, mimicry provides functions of social regulation. Importantly, a positive affiliation between observer and target is suggested to be a necessary requirement for emotional mimicry to occur, especially if the mimicked expression is of negative valence[21]. To illustrate, the context-free reproduction of a nemesis's frown would be counter-intuitive, given that their demise would induce satisfaction rather than sadness. As such, emotional mimicry necessitates a positive affiliation or the explicit motivation to infer another's affective state[20].

Investigating mimicry in empathy-inducing situations, Lamm and colleagues[19] instructed participants to watch video clips of patients receiving painful sonar treatment while either imagining the patients' feelings or receiving the treatment themselves. Corrugator supercilii (frowning muscle) activity measured via electromyography (EMG) increased irrespective of the taken perspective, while the activity of the orbicularis oculi (lid tightening muscle) only rose if participants imagined themselves in the same situation. Murata et al.[20] corroborated these findings by revealing a higher tendency to mimic facial expressions when participants were directly asked to infer another's affective state, both in terms of activity in four different muscle sites and regarding distinct facial expressions rated using the facial action coding system (FACS)[22]. The FACS constitutes a taxonomy to categorize movements of facial muscles or muscle groups in terms of action units (AUs). For example, AU12 describes the action of lip corner pulling, a facial movement governed by the zygomaticus major muscle.

In one of the few studies investigating emotional mimicry in a real-life dyadic setting, Riehle and colleagues[23] measured the EMG activity of the corrugator supercilii and the zygomaticus major (smiling muscle) in a dyadic discussion task. Using windowed-cross-lagged-correlation[24], the authors detected significant covariation only for zygomaticus activity. These findings inform research in the context of psychosocial stress. Both, emotional mimicry occurring when perspective taking is manipulated[19,20] and a lack of synchronization for negatively valanced facial behavior between strangers[23], are in line with previous findings regarding lower stress transmission between strangers in comparison to romantic couples[9] and higher trait cognitive empathy associating with greater empathic stress occurrence in observers[9,13].

In the current study, we investigated subjective-psychological and physiological empathic stress and its association with emotional mimicry in adolescents observing either their mothers or fathers undergo the TSST. Considering the multi-modality and complexity of the human stress response[25], adolescents and parents simultaneously provided measurements of cortisol release, sympathetic, parasympathetic, and subjective stress reactivity. Additionally, dyads were videotaped during the stress task to gauge adolescents' emotional mimicry via shared AU activity. Based on the notion that mimicry acts as a sub-conscious precursor of affective and cognitive empathic processes[2], we expected greater adolescent stress-resonance and vicarious stress gave a higher degree of emotional mimicry between parents and adolescents.

## Methods
### Transparency and openness
Hypotheses and testing procedures of this study were preregistered on 16 May 2022, prior to data collection, and can be accessed at https://osf.io/3gdym. Any deviations from the preregistration are flagged in the Methods. The preregistered methods were updated during our piloting sessions (before official data collection) because the previously planned recording of electromyography was incompatible with proper mimicry assessments. Due to the extent of the current analyses, the preregistered hypotheses concerning adolescents' affective and cognitive empathic abilities were moved to a second manuscript. Analysis scripts and data are publicly accessible at https://osf.io/qg9yc/. All materials used in this study are widely available. No artificial intelligence-assisted technologies were used in this research or the creation of this article.

### Participants
Our sample consisted of $N = 77$ parents (41 women, 36 men, age $M = 44.9$, SD = 5.09, age range = 35–62) accompanied by their adolescent children (40 girls; 37 boys; age $M = 14.2$, SD = 1.06, age range = 13–16). No information on race/ethnicity was collected. Adolescents between the ages of 13 and 16 were chosen to ensure sufficient understanding of the stressful aspects of the TSST, which was likely lacking in a previous study in a younger child cohort between the ages of 8–12[13]. Participation in the study was promoted in and around the city of Jena. Before being invited to the laboratory, parents completed a standardized telephone interview assessing inclusion and exclusion criteria. These criteria were chosen to ensure that cortisol measurements were unconfounded and participants could be safely confronted with a psychosocial stress test. Parent-adolescent dyads were excluded if mothers or fathers were strong smokers (>10 cigarettes a day), reported recreational drug consumption, were unwilling to abstain from alcohol intake for at least a week, had a BMI < 18.5 or >30, had previously completed a standardized laboratory stressor, or if either parents or adolescents reported dyslexia or non-fluency in German.

In terms of current physiological and psychological health, dyads were excluded in case of significant health problems, recent stressful life events such as separation or death of a partner/parent, ongoing psychotherapy, diagnosed mental disorders in the last 2 years, or usage of medication affecting HPA axis activity (e.g. steroids). Daughters were excluded from the study if they reported usage of hormonal contraceptives or were pregnant. At the beginning of data collection, mothers were also excluded if they

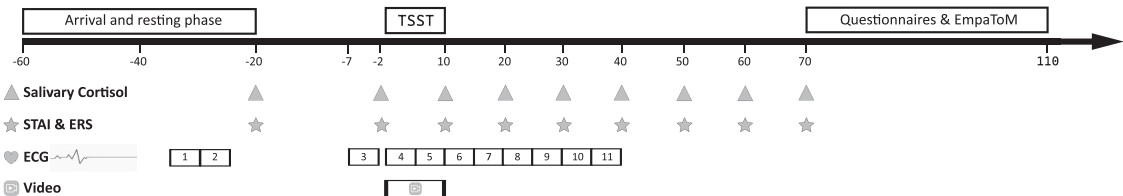

**Fig. 1 | Testing timeline.** Parents and their adolescent children were brought to the waiting room at −60 min before stressor onset. Salivary cortisol and subjective stress samples were collected at −20, −2, +10, +20, +30, +40, +50, +60, and +70 min. Adolescents also repeatedly reported on their state empathic concern and personal distress. Furthermore, an ECG recording was sampled from −40 to +42 min. The recording was split into eleven 5-min timeframes (1–11) and heart rate and HRV were averaged for parents and adolescents for each timeframe.

reported usage of hormonal contraceptives, were pregnant, or in menopause. However, these restrictions were dropped due to sampling difficulties, which led to the inclusion of 12 mothers (29.3% of mothers) not on a natural cycle. In detail, four mothers reported using a hormonal coil (9.76%), four mothers reported being in menopause (9.76%) and four mothers reported using oral contraceptives (9.76%).

Controlling for the menstrual cycle phase in both mothers and daughters (i.e. testing during the luteal phase, during which cortisol release in females is most similar to males[26]) was unfeasible because daughters and their mothers would only show a synchronized menstrual cycle by chance. Thus, we tested six mothers during their period (14.6%), 12 mothers during their follicular phase (29.3%) and 11 mothers during their luteal phase (26.8%). Regarding daughters, seven were tested during their period (17.5% of daughters), seven during their follicular phase (17.5%), 15 during their luteal phase (37.5%), and three before the onset of menarche (7.5%). Eight daughters reported that they were unsure of their current menstrual cycle phase (20%). None of the daughters used hormonal contraceptives.

The study was approved by the Research Ethics Board of Jena University Hospital (ethics number: 2019-1578). Parents and children provided written informed consent for both themselves and their children and were financially compensated for their time and effort. Participants were informed that they could withdraw from the study at any point in time.

### Experimental design and procedure

Data collection took place at the Institute for Psychosocial Medicine, Psychooncology and Psychotherapy in Jena, between July 2022 and March 2024. All testing sessions were scheduled on weekdays and between 12 p.m. and 7 p.m. to control for the cortisol circadian rhythm[27]. After arriving at the laboratory, parents and adolescents were brought to the waiting room and offered a glass of juice and a chocolate bar to equalize blood sugar levels. Afterward, only water was available for the remainder of the testing session. Deviating from earlier empathic stress studies[9,13], and with the aim to maximize parent–adolescent covariance, dyads were not separated upon arrival but spent the entire testing session together. Dyads rested for 40 min (baseline phase) to mitigate the potential stress triggered by the unfamiliar laboratory situation. At the end of the baseline phase, baseline measurements of salivary cortisol and subjective stress were collected in parents and adolescents (at −20 min relative to the onset of the TSST at 0 min). At this time, adolescents also provided their first sample of self-reported state empathic concern and state personal stress. Fifteen minutes before stressor onset (−15 min), adolescents were brought to the testing room and seated next to the committee administering the TSST, with their parents following shortly afterward. Adolescents were asked to act strictly as passive and silent observers. As such, they were instructed to make no direct contact with either their parents or the TSST committee members. Dyads provided the second and third measurements of cortisol and subjective stress after the 5-min stress *anticipation phase* (−2 min before stressor onset), and immediately after the TSST (+10 min after stressor onset). Additionally, from 0 to +10 min, facial videos were recorded by both parents and adolescents to assess facial action unit (AU) activity (see Fig. 1 for the testing timeline). Subsequently, dyads were brought back to their waiting room where they spent the remainder of the testing session together. During this time, cortisol and subjective stress were sampled at +20, +30. +40, +50, +60 and +70 min, comprising the *recovery phase* (+20–+70 min). Before and after the TSST, dyads were allowed to freely interact with each other. They were encouraged to bring their own reading material as a pastime, except for material related to work or school. Participants were asked to switch off their smartphones during the entirety of the testing session.

Across the testing session, a continuous electrocardiogram (ECG) was recorded to assess sympathetic and parasympathetic activity (from −40 until +42 min). Towards the end of the testing session (starting at +70 min), participants also completed a set of questionnaires (data not subject to the current study). After the recovery phase, a computerized empathy task, the EmpaToM[28], was completed (data not subject to the current study).

### Trier social stress test

Parents completed the Trier social stress test (TSST[15], a standardized psychosocial laboratory stressor. The TSST consists of an anticipation phase (5 min), a mock-job interview (5 min) and a mental arithmetic task (5 min), all of which are completed in front of a gender-mixed committee. The gender-mixed committee is introduced as trained behavioral analysts and instructed to show no reaction to the performance of the stressed targets. The TSST has been shown to reliably elicit a stress response in about 80% of adult participants[15,29].

### Measures of acute stress reactivity

**Cortisol**. To capture the activity of the HPA axis, salivary cortisol was sampled using Salivettes (Sarstedt, Nümbrecht, Germany). During sampling, a saliva collection swab was kept in the mouth for 2 min and participants were asked to refrain from chewing. Until analysis at the biochemical laboratory of the Biological and Clinical Psychology Department of Trier University, Salivettes were stored at −30 °C. A time-resolved fluorescence immunoassay with intra- and interassay variabilities of less than 10% and 12% was used to assess cortisol activity (nmol/l)[30].

**Autonomic activity**. ECG data was recorded in parents and adolescents using a Zephyr Bioharness 3 chest belt (Zephyr Technology, Annapolis, MD USA) at 250 Hz for 82 min (from −40 to +45 min). Activity of the sympathetic nervous system in association with the parasympathetic nervous system was operationalized via heart rate[31]. Activity of the parasympathetic and sympathetic nervous systems was assessed using the root mean square of successive differences (RMSSD; Berntson et al.[32]). This measure was chosen because breathing frequency varies throughout childhood and adolescence, making it incomparable to adult respiratory rates[33]. Since HF-HRV reflects the high-frequency domain of HRV (0.15–0.4 Hz), which is primarily influenced by respiratory sinus arrhythmia and thus breathing rate[32], adolescents' higher breathing rates would necessitate a different high-frequency range. Thus, we chose RMSSD as a measure of HRV because it is not dependent on respiratory frequency.

To link specific phases of the testing protocol to the ECG recording, overall eleven 5-min timeframes were extracted (Fig. 1). Two baselines

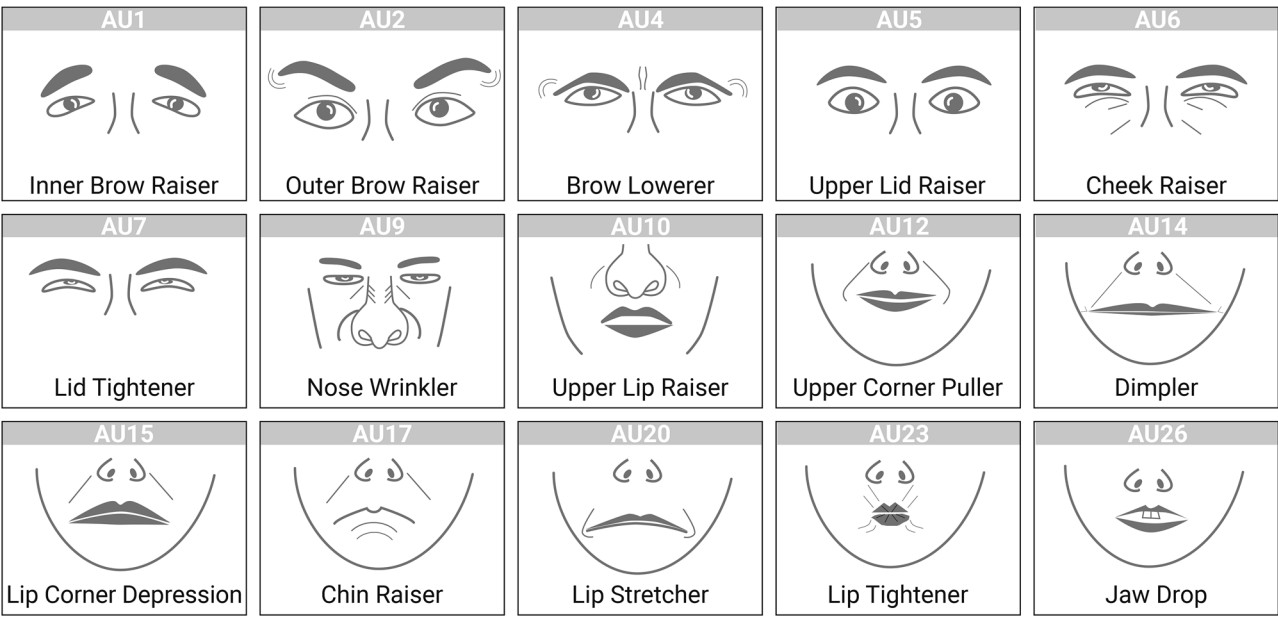

**Fig. 2 | Action units.** Action units extracted by open face. AU25 (lips part) is not depicted due to its high similarity to AU26.

phases (from −35 to −25 min) represented the time before participants entered the testing room. Time phase 3 covered the 5 min of the anticipation phase (from −7 to −2 min) while time phases 4 and 5 covered active stress induction (from 0 to +10 min). Time phase 6 started approximately two minutes after termination of the stress phase (from +12 to +17 min) to account for the time it took participants to return to the waiting room. For the recovery phase, time phases 6, 7, 8, 9, 10, and 11 reflected the remaining 30 min of the ECG recording (from +12 to +42 min).

Raw ECG recordings were manually checked for artifacts (e.g. ectopic beats, signal loss) by two independent research assistants using Python-based in-house software. If heartbeats were impossible to extract for a given portion of the ECG recording, the recording was cut. If 10% or more of a given time phase was cut, the time phase was dropped from further analysis. Average heart rate (bpm) and RMSSD (ms) were calculated for each 5-min time phase using the hrv-analysis package[34].

**Subjective stress and state empathy.** Self-reported stress was measured throughout the testing session using the 20-item state scale of the state-trait anxiety inventory (STAI)[35]. The STAI prompts feelings of tension, nervousness, apprehension, and arousal on a scale from 1 (not at all) to 4 (very much).

Only in adolescents, state empathic concern and personal distress were sampled with the 14-item empathic response scale (ERS)[36] simultaneously with salivary cortisol and subjective stress. The ERS prompts feelings for another individual on six adjectives measuring empathic concern (e.g. "softhearted", "compassionate") and eight adjectives measuring personal distress (e.g. "troubled", "worried") on a seven-point Likert scale from 1 (not at all) to 7 (very much).

**Action unit activity**
During the stress phase of the TSST (from 0 to 10 min), both parents' and adolescents' upper torsos and faces were videotaped. Panasonic Full HD-Camcorder HC-V180 video cameras were used, filming at 25 frames per second resulting in $N = 15,000$ frames per participant. Video angle was adjusted for participants in terms of height differences. Lighting in the testing room was held constant by only using artificial light and opaque window blinds. Videos of parents and adolescents were synced via audio waveform synchronization.

To assess action unit (AU) activity, videos were analyzed using OpenFace[37], an open-source facial expression toolkit. OpenFace provides frame-by-frame measures of confidence (subject face visible vs. obstructed), AU presence (not active vs. active), and continuous AU intensity ranging from 0 (no activity) to 5 (high intensity) for 16 AUs (see Fig. 2). Thus, for each action unit, a time series consisting of 15,000 measurement time points was extracted for both parents and adolescents. For an overview of parents' and adolescents' mean AU activity and occurrence rates, please see Figs. S3 and S4 in the Supplementary Information.

**Statistical analysis**
Analyses were performed in R 4.2.2[38].

**Stress marker data preparation.** RMSSD HRV data were transformed via the natural logarithm because subsequent model residuals were non-normally distributed. To attain a measure of parental and adolescent stress sensitivity, the area under the curve with respect to increase (AUCi)[39] was calculated for both dyad members and each stress marker (cortisol, heart rate, HRV, subjective stress). Because state empathic concern and state personal distress were only measured in adolescents, AUCis were only calculated for them. Contrasting our data analysis preregistration, stress markers were not universally log-transformed and winsorized to 2SD in response to a reviewer's suggestion.

**Manipulation check.** To confirm successful stress induction in parents, significant cortisol responder rates were determined based on the calculated cortisol change scores. A cortisol increase of at least 1.5 nmol/l has been suggested to reflect a significant stress response[16].

**Empathic stress responders.** As in previous studies[9,13,40,41], we also determined empathic stress responder rates among observers using the 1.5 nmol/l criterion[16].

**Overview mimicry calculation.** To quantify differences in the amount of mimicry shown by adolescents, we employed a four-step procedure to acquire a measure of the *duration of time spent synchronized* for each dyad on each AU. First, we calculated *windowed-cross-lagged-regressions* (WCLR)[42] across all sixteen parent-adolescent AU intensity time series. In a second step, we calculated *sync-intervals* in each of the sixteen WCLR matrices per dyad[43]. The lengths of the sync intervals were summed up to obtain the duration dyads spent synchronized on a given AU. Third, an explorative factor analysis (EFA) was employed to calculate a structural

equation model (SEM) across the 16 synchronization durations. Lastly, latent factor scores were extracted from the SEM for each dyad. We used an SEM approach because synchronization durations showed significant correlations between AUs (see Fig. S2 in the Supplementary Information) due to the simultaneous activity of different facial muscles in one and the same emotional expression (e.g. when smiling, AU06, AU07, and AU12 are activated simultaneously).

**WCLR.** WCLR has been previously employed in the investigation of movement synchrony[44]. A highly similar approach using correlations (windowed-cross-lagged-correlation) has been employed to examine dyadic mimicry[23,45]. WCLC and WCLR are methods to assess the degree of association between two time series at each measurement time point, while also taking into consideration different time lags. In other words, the synchrony of two vectors is calculated simultaneously (person A and person B smile at the exact same time) and at different time lags (person A smiles and person B smiles 1 s later). Because in our study adolescents were observing their parents, synchronization was only investigated in one direction. Although parents were also able to see their children and could have synchronized with their expressions, our particular interest was the degree to which adolescents mimicked their parents' facial activity. Consequently, a one-sided WCLR was performed with lagged linear regressions calculated for the adolescents lagging *behind* their parents. To facilitate understanding of the WCLR approach, we created a video demonstration accessible at https://osf.io/spvk8.

To calculate the WCLR, a constant *window-size* must be specified, determining the number of observations from the full-time series that are regressed at each time point. At a given window position, regressions are calculated at different time lags. The *maximum lag* is defined by how much later the synchronized behavior is expected to be shown. The *lag increment* determines how many different lags are assessed before the maximum lag is reached. The window is then moved across the full time series by a *window-increment* and again, cross-lagged regressions are calculated for the next window increment.

Given a certain window at a specific lag (e.g. window = 1–125, lag = 2), WCLR then compares whether the amount of variance explained by the partner (i.e., the parent) exceeds that of an auto-regression (i.e., the adolescent's previous activity). Thus, auto-correlation existent in the lagged signal is controlled for. In detail, two models are fitted and compared at each step [ref. 42, p. 3].

$$\text{Model 1: } X_{1t+\tau} = \beta_0 + \beta_1 X_{1t} + \varepsilon_{1t} \tag{1}$$

$$\text{Model 2: } X_{1t+\tau} = \beta_0 + \beta_1 X_{1t} + \beta_2 X_{2t} + \varepsilon_{1t} \tag{2}$$

In our case, $X_{1t}$ pertains to the data points of the adolescent and $X_{2t}$ pertains to the data points of the parent at a time t. Model 1 is a simple auto-regressive model, in which the adolescent's data points $\tau$ steps further ($X_{1t+\tau}$) are explained by their initial data points ($\beta_1 X_{1t}$).

Model 2 includes the auto-regressive effect ($\beta_1 X_{1t}$) and the effect of the parents ($\beta_2 X_{2t}$).

To gauge an index of cross-correlation irrespective of autocorrelation, $R^2_{cc}$ is calculated as

$$R^2_{cc} = R^2_{\text{Model 2}} - R^2_{\text{Model 1}} \tag{3}$$

To further assure that the cross-correlation is not random, the $R^2_{cc}$ is only retained if model 2 explains significantly more variance than model 1. For a more detailed description of WCLR, please see the relevant literature[42–44].

Before WCLR was employed, all time-series were log-transformed via the natural logarithm and smoothed using a moving average [$n = 10$][44]. Missing values were replaced by random numbers between 0 and 1.79 (uniform random distribution). This was done to ensure that synchrony was not accidentally introduced if one or both dyad members showed long strings of missing values (i.e., more than 10 consecutive missing values). We

added Gaussian noise to the time series (SD = 0.01) to make regression modeling possible for longer intervals of non-activity (zeros).

For our WCLR analysis, we chose a 5 s window size (125 frames) with a window increment of 1/25 s (1 frame), a maximum lag of 2 s (50 frames), and a lag increment of 1/25 s (1 frame). We chose similar parameters as Riehle et al.[23], the current single study investigating facial mimicry using WCLC. In detail, we expected the behavior of interest (changes in AU intensity) to be at most 5 s (125 frames) long, and subsequent mimicry in adolescents to reveal itself at the latest 2 s following the behavior of interest (50 frames). With a video length of 10 min at 25 frames per second (15,000), this resulted in a matrix of up to 756,075 $R_{cc}^2$ for each AU per dyad (14,825 windows with 51 lags each). The total number of windows is calculated by subtracting the length of each window (125) and the maximum lag (50) from the total length of the time series (15,000), resulting in 14,825 windows. This way, each $R^2$ calculated includes exactly 125 observations (windows size) per dyad member. Each window includes 51 possible lags because the first lag (i.e. no lag) is included. Subsequently, all negative slopes and slopes with an $R_{CC}^2 < 0.25$ were dropped from the $R_{CC}^2$ matrices. While negative slopes were dropped because they would index counter-mimicry rather than emotional mimicry, $R_{CC}^2 < 0.25$ acted as a threshold to limit spurious correlations as suggested by Schoenherr et al.[44]. As a rather conservative effect-size cut-off, $R_{CC}^2 < 0.25$ should be a valid multiple testing approach as well, given the number of regressions tested (756,075).

**Sync intervals.** To calculate the time a dyad was synchronized for each AU, we summed up the lengths of the sync-intervals that were present in each WCLR matrix. Sync intervals are comprised of peak $R_{CC}^2$ in subsequent windows at the same lag[43]. For example, if a dyad showed a peak $R_{CC}^2$ at a lag of 25 between windows number 100 and 200 on AU12, the adolescent was synchronizing with the parent after one second (lag of 25 frames) for 4 s (interval length of 100 frames). Another short video demonstration for a better understanding of sync interval calculation is accessible at https://osf.io/fcg6a.

First, all local maxima were extracted in each of the 14,825 windows of a given WCLR matrix, resulting in a number of peak $R_{cc}^2$ at a specific lag for each window. Peaks were defined as the highest value between two ascending and two descending values. Second, sync intervals were extracted by checking for peak neighbors. We defined peak neighbors as peaks on adjacent windows and at a lag difference of at most ±1[43].

The length of the sync intervals was determined by counting the length of each series of peak neighbors. Sync intervals shorter than 10 peaks (0.4 s) were excluded as suggested by Schoenherr et al.[46]. If sync intervals were overlapping, that is, two peak series were present on the same (or partly the same) series of windows, the sync interval with the lower mean $R_{CC}^2$ was dropped. Finally, for each dyad and for each AU, the relative amount of *time spent synchronized* was calculated by dividing the total amount of sync time (amount of peaks in all sync intervals) by the total amount of time (10 min or 15,000 measurement time points). In other words, a percentage of time synchronized was extracted for each of the 16 AUs per dyad. Importantly, this metric only indirectly gauges the strength of synchronization. While sync intervals will include comparatively high correlations due to the nature of peak-picking, a sync interval does not necessarily entail an intense facial expression in both the target and observer.

Mean sync time and the resulting mean $R_{cc}^2$ and lag for each AU can be found in Table 1. The average lag of the respective sync intervals calculated for each AU is depicted in Fig. S6 in the Supplementary Information.

**Exploratory factor analysis.** Because sync times across the 16 AUs correlated across dyads, we chose an SEM approach to model possible latent factors driving the covariation between AU sync times. The number of factors in our EFA was calculated using parallel analysis[47] and diagnosed with a scree-plot. In parallel analysis, the eigenvalues of the factor solution are compared to the eigenvalues generated via Monte-Carlo simulation using random data (our parallel analysis scree-plot is shown in the Supplementary Information; Fig. S1). Subsequently, EFA

**Table 1 | Average sync times and regression slopes for each AU**

| AU | Mean sync time in % | Mean $R^2_{CC}$ | Mean lag |
|---|---|---|---|
| AU01 | 16.11 (3.61) | 0.47 (0.03) | 32.77 (2.27) |
| AU02 | 9.50 (3.14) | 0.47 (0.04) | 32.69 (2.79) |
| AU04 | 14.85 (5.60) | 0.43 (0.03) | 33.28 (2.08) |
| AU05 | 9.04 (2.70) | 0.47 (0.04) | 31.07 (3.11) |
| AU06 | 11.70 (5.70) | 0.45 (0.04) | 33.54 (4.95) |
| AU07 | 17.73 (6.23) | 0.43 (0.03) | 32.19 (2.34) |
| AU09 | 7.33 (2.82) | 0.41 (0.03) | 30.57 (3.59) |
| AU10 | 17.10 (5.09) | 0.44 (0.03) | 32.87 (2.52) |
| AU12 | 13.04 (5.35) | 0.46 (0.04) | 34.66 (2.63) |
| AU14 | 17.59 (4.96) | 0.44 (0.02) | 33.87 (2.36) |
| AU15 | 16.20 (3.63) | 0.44 (0.02) | 32.28 (2.44) |
| AU17 | 20.05 (3.50) | 0.42 (0.02) | 33.02 (2.07) |
| AU20 | 14.28 (3.41) | 0.44 (0.03) | 31.61 (2.48) |
| AU23 | 12.51 (3.38) | 0.48 (0.03) | 32.99 (2.66) |
| AU25 | 17.92 (3.57) | 0.42 (0.02) | 32.23 (2.15) |
| AU26 | 18.69 (2.30) | 0.42 (0.02) | 32.68 (2.09) |

Standard deviation in brackets. Sync time refers to the amount of time dyads were synced on a given AU, as calculated via WCLR and sync intervals. Mean $R^2_{CC}$ were calculated after sync interval detection, meaning that all $R^2_{CC} < 0.25$ were already dropped.

was employed using the oblimin rotation method for a two-factor solution. In the last step, two latent scores were derived from the final two-factor SEM model for each dyad, reflecting their respective sync time on two mimicry latent factors.

**Regression modeling.** For each stress marker, we calculated a multiple linear regression model predicting adolescent AUCi by parent AUCi as well as two-parent AUCi by mimicry-factor interactions. While the main effects on the intercept in this model can be interpreted as effects on vicarious stress reactivity (i.e., effects that are independent of parental reactivity), interactions with parental AUCi can be interpreted as effects on stress-resonance. Regression estimates (Est.), confidence intervals (CI), $p$-values, and Bayes factors (BF) are reported. BF was calculated in comparison to the null model (intercept only). A BF $\leq 0.33$ suggests credible evidence for the null-hypothesis, while a BF between 0.34 and 1 suggests anecdotal evidence. Adolescent AUCi and the mimicry factors were $z$-standardized to minimize multicollinearity. Please see Table S2 in the Supplementary Information for the full regression tables.

**Sensitivity analysis.** Because we preregistered the conduction of WCLC and not WCLR, we conducted a sensitivity analysis, mirroring the WCLR approach. In other words, window-size, maximum lag, window increment, and lag increment were held constant. Sensitivity analysis results lead to almost identical latent factor structure and regression modeling results.

**Multiple testing.** Because our hypotheses regarding the effects of mimicry and socio-emotional capacities on empathic stress were tested across overall six stress markers, a Bonferroni correction was applied. The significance threshold was adjusted to $p = 0.0167$ (0.05/3) reflecting two groups of three physiological (cortisol, HR, HRV) and three psychological (STAI, EC, PD) stress markers. Previous studies have shown little to no covariance between physiological and subjective stress reactivity, illustrating the complexity of the human stress response[25,48]. Because physiological or psychological empathic stress is thus interpreted as a specific rather than general case of empathic stress, no further correction was applied[49]. This approach to control for multiple testing was not

included in our preregistration but was later implemented in response to a reviewer request.

**Missing data.** Because AUCi calculation requires full data sets, up to two cortisol, STAI, empathic concern, and personal distress values were imputed via multiple imputations using the mice package[50]. For heart rate and HRV, up to three values were imputed, because 11 time-phases were extracted from the ECG recordings in comparison to nine cortisol and STAI measurements. Percentages of values imputed for each stress marker can be found in Table S1 of the Supplementary Information.

Adolescents were excluded from analysis via row-wise deletion because they or their parents had no video data or because AUCi could not be calculated due to excessive missing values. Six dyads were excluded because WCLR could not be calculated due to missing video data. Regarding stress data, one further dyad was excluded from cortisol analysis ($N = 70$) as well as STAI analysis ($N = 70$), and two dyads were excluded from state personal distress analysis ($N = 71$). Unfortunately, adolescents revealed an excessive amount of heart rate and HRV missings, leading to the exclusion of 34 dyads from analysis due to missing more than three time phases, resulting in a heavily reduced data set of $N = 40$ dyads. An overview of the observations included in the regression analysis can be found in Table S1 of the Supplementary Information.

**A-priori power analysis.** For a one-sided t-test of a single regression coefficient in our multiple linear regression models with six predictors each (parent-AUCi, mimicry, an interaction, puberty status, hormonal status, and dyad type), expecting an effect size of at least $f^2 = 0.10$[13], at an alpha level of 0.05 and a power level of 0.80, a minimum sample size of $N = 64$ dyads was required.

**Sensitivity power analysis.** The final cortisol and STAI models included $N = 70$ dyads. A sensitivity power analysis revealed the sample size has 80% power to detect an effect size of at least $f^2 = 0.09$.

The final heart rate and HRV models included $N = 37$ dyads. A sensitivity power analysis revealed the sample size has 80% power to detect an effect size of at least $f^2 = 0.17$.

The final empathic concern model included $N = 71$ dyads. A sensitivity power analysis revealed the sample size has 80% power to detect an effect size of at least $f^2 = 0.09$.

The final personal distress model included $N = 69$ dyads. A sensitivity power analysis revealed the sample size has 80% power to detect an effect size of at least $f^2 = 0.09$.

**Reporting summary**

Further information on research design is available in the Nature Portfolio Reporting Summary linked to this article.

## Results

### Manipulation check

A total of 60 (77.9%) parents exhibited a cortisol stress response of >1.5 nmol/l from baseline to their individual cortisol peak, suggesting successful stress induction[16]. $N = 26$ (63.3%) mothers and $N = 33$ (94.4%) fathers reached the >1.5 nmol/l threshold (see Fig. 3 for an overview of the stress marker trajectories).

### Empathic stress responders

$N = 9$ (11.7%) adolescents showed significant cortisol release of >1.5 nmol/l. In detail, $N = 5$ daughters and $N = 4$ sons were categorized as cortisol responders. Regarding the type of parent that was watched, $N = 7$ or 17.1% of adolescents watching their mother and $N = 2$ or 5.55% of adolescents watching their fathers were categorized as cortisol responders.

### Sync time across action units

On average, adolescents were synced with their parents' AU activity for 14.6% (2.416 min) of the 10-min TSST duration (SD = 2.00). Only AU02,

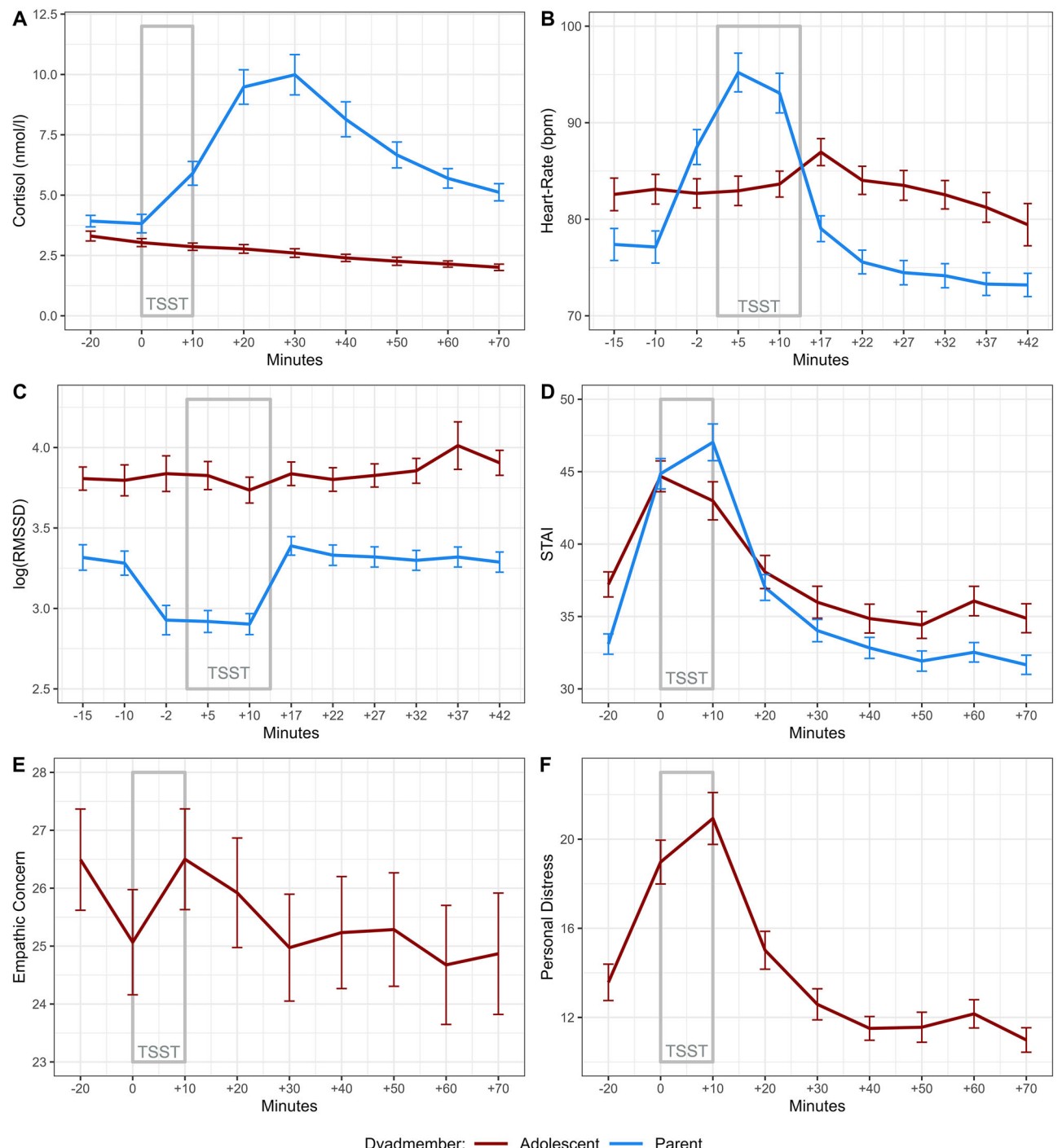

**Fig. 3 | Parent and adolescent stress marker trajectories.** Trajectories of stress markers for adolescents (red) and parents (blue) in terms of cortisol **A** with $n = 70$ dyads included,, STAI **B** with $n = 70$ dyads included, heart rate **C** with $n = 37$ dyads included, HRV **D** with $n = 37$ dyads included, empathic concern **E** with $n = 71$ dyads included and state personal distress state **F** with $n = 69$ dyads included across the testing session. Standard errors are depicted for each measurement point.

AU05 and AU09 were synchronized for <10% of the total TSST time. The average sync time for each AU can be found in Table 1.

On average, none of the AUs were synced for more than 30% of the time (see Fig. 4). Mother–daughter dyads were synced the longest ($M = 15.4$, SD = 2.08), followed by mother–son dyads ($M = 14.7$, SD = 1.76), father–son dyads ($M = 14.4$, SD = 2.29) and father–daughter dyads ($M = 13.7$, SD = 1.50). These differences were not significant ($F$ (3,67) = 2.39, $p = 0.076$, $\eta^2 = 0.097$, CI [0.004, 0.212]).

Sync intervals had an average lag of $M = 32.68$ (SD = 0.84), meaning that, on average, adolescents synced with their parents AU activity after 1.31

(SD = 0.03) s. An overview of the average lag for each AU can be found in the Supplementary Information section (Fig. S5).

Correlations between sync times can be viewed in the Supplementary Information (Fig. S2).

**Structural equation model**
Parallel analysis revealed a two-factor solution to best fit the correlational structure of the 16 synchronization durations per dyad (see Fig. S1). The two-factor model Chi-Square test was not significant ($\chi^2(53) = 53.01$, $p = 0.474$). The first latent factor (lf1) loaded onto AU01, AU02, AU09,

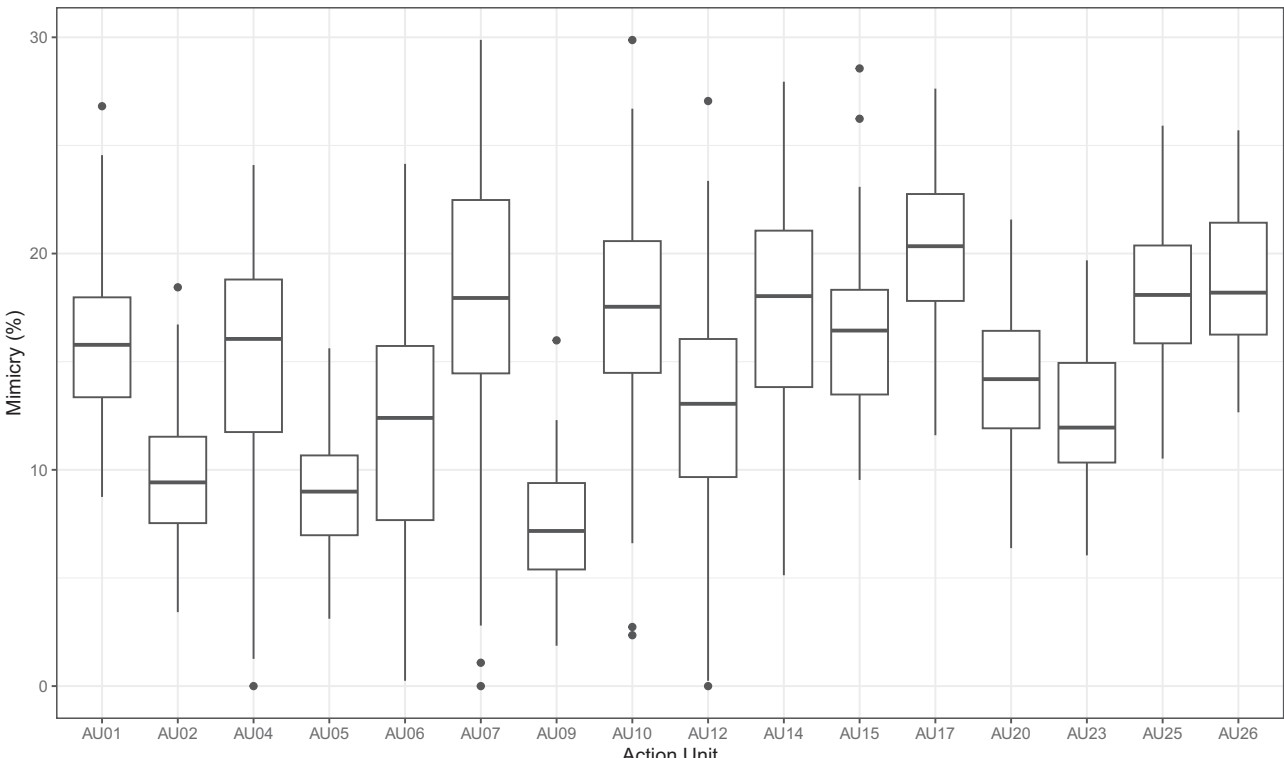

**Fig. 4 | Average time spent synchronized per action unit across all dyads (in percent).** $n = 71$ dyads included. Means (horizontal line in the boxplot), the 25th and 75th percentiles (hinges) and outliers of time spent synchronized for each action unit. Percentages are in relation to a maximum of 10 min (100%) of time spent synchronized.

AU14, AU15, AU17, AU20, AU23, AU25 and AU26 sync times. The second latent factor (lf2) loaded onto AU06, AU07, AU10, and AU12 sync times (see Fig. 5). Thus, the model revealed a first, complex factor that subsumed synchronizations on AUs coding for various negative emotions, including sadness (AU01, AU04, AU15), fear (AU01, AU02, AU04, AU05, AU07, AU20, AU26), anger (AU04, AU05, AU07, AU23) disgust (AU09, AU15, AU16), and surprise (AU01, AU02, AU05, AU26)[22]. The second factor subsumed synchronizations on AUs coded as the act of smiling or contentment (AU06, AU07, AU10, AU12, AU14). Hence, the first factor can be understood as synchronization with negative expressions and the second factor can be understood as synchronization with positive expressions. None of the factors loaded onto AU04 or AU05 sync times.

### Influences of mimicry on empathic stress

To investigate the associations of mimicry with empathic stress, multiple linear regression models were calculated for each stress marker. Adolescent AUCi was predicted by parental AUCi, the two latent scores (negative vs. positive) extracted from the SEM, and interactions between parental AUCi and the two latent scores. While main effects on the intercept can be interpreted as effects on vicarious stress reactivity (independent of parental stress reactivity), interactions with parental AUCi can be interpreted as effects on stress-resonance.

**Cortisol**. Cortisol trajectories showed a decrease rather than an increase across the testing session, illustrated by 55.9% of adolescents showing their highest individual cortisol measurement at baseline (see Fig. 3). There was anecdotal evidence for a null-effect of the negative (Est. = −3.48, CI [−11.63, 4.67], $p = 0.397$, BF = 0.34) and credible evidence for a null-effect of the positive (Est. = 0.54, CI [−7.61, 8.69], $p = 0.895$, BF = 0.25) latent mimicry scores on adolescent cortisol AUCi. Regarding the main effect of parental on adolescent cortisol AUCi, we found credible evidence for a lack of cortisol stress-resonance (Est. = −2.13, CI [−10.05, 5.80], $p = 0.594$, BF = 0.26). Additionally, there was anecdotal

evidence for a lack of moderation by the negative (Est. = −0.54, CI [−9.19, 8.11], $p = 0.901$, BF = 0.59) and credible evidence for a lack of moderation by the positive (Est. = 7.05, CI [−0.98, 15.08], $p = 0.084$, BF = 0.15) latent mimicry scores.

**Heart rate**. The majority of adolescents peaked immediately after the TSST (60.0%). There was anecdotal evidence for a null-effect of the negative (Est. = 91.09, CI [−95.43, 277.62], $p = 0.327$, BF = 0.55) and credible evidence for a null-effect of the positive (Est. = −45.45, CI [−212.42, 121.51], $p = 0.583$, BF = 0.33) latent mimicry scores on adolescent heart rate AUCi. Regarding the main effect of parental on adolescent heart rate AUCi, we found anecdotal evidence for a lack of heart rate stress-resonance (Est. = 241.69, CI [−15.82, 499.19], $p = 0.065$, BF = 0.76). Additionally, there was anecdotal evidence for a lack of moderation by the negative (Est. = −109.25, CI [−319.18, 100.69], $p = 0.297$, BF = 0.39) and credible evidence for a lack of moderation by the positive (Est. = 137.67, CI [−126.82, 402.15], $p = 0.297$, BF = 0.19) latent mimicry scores.

**HRV**. In contrast to the other stress markers, HRV decreases with greater stress. Thus, low or negative AUCi values index greater stress reactivity. The majority of adolescents reached their nadir right before the TSST (24.4%), in the first phase of the TSST (17.8%) or immediately after the TSST (26.7%). There was anecdotal evidence for a null-effect of the negative (Est. = −1.25, CI [−12.15, 9.65], $p = 0.817$, BF = 0.66) and credible evidence for a null-effect of the positive (Est. = 3.93, CI [−5.57, 13.42], $p = 0.406$, BF = 0.32) latent mimicry scores on adolescent HRV AUCi. Regarding the main effect of parental on adolescent HRV AUCi, we found anecdotal evidence for a lack of HRV stress-resonance (Est. = 11.83, CI [−3.04, 26.71], $p = 0.115$, BF = 0.55). Additionally, there was anecdotal evidence for a moderation by the negative (Est. = −14.43, CI [−27.11, −1.76], $p = 0.027$, BF = 1.63) and credible evidence for a lack of moderation by the positive (Est. = −4.90, CI [−22.42, 12.62], $p = 0.572$, BF = 0.19) latent mimicry scores.

**Fig. 5 | SEM factor solution for action unit sync times across participants.** The EFA revealed a two-factor solution. The first latent factor (lf1) loaded onto AU01, AU02, AU09, AU14, AU15, AU17, AU20, AU23, AU25 and AU26 (various negative emotions). The second latent factor (ldf2) loaded onto AU06, AU07, AU10 and AU12 (smiling or contempt). None of the factors loaded onto AU04 or AU05 sync times.

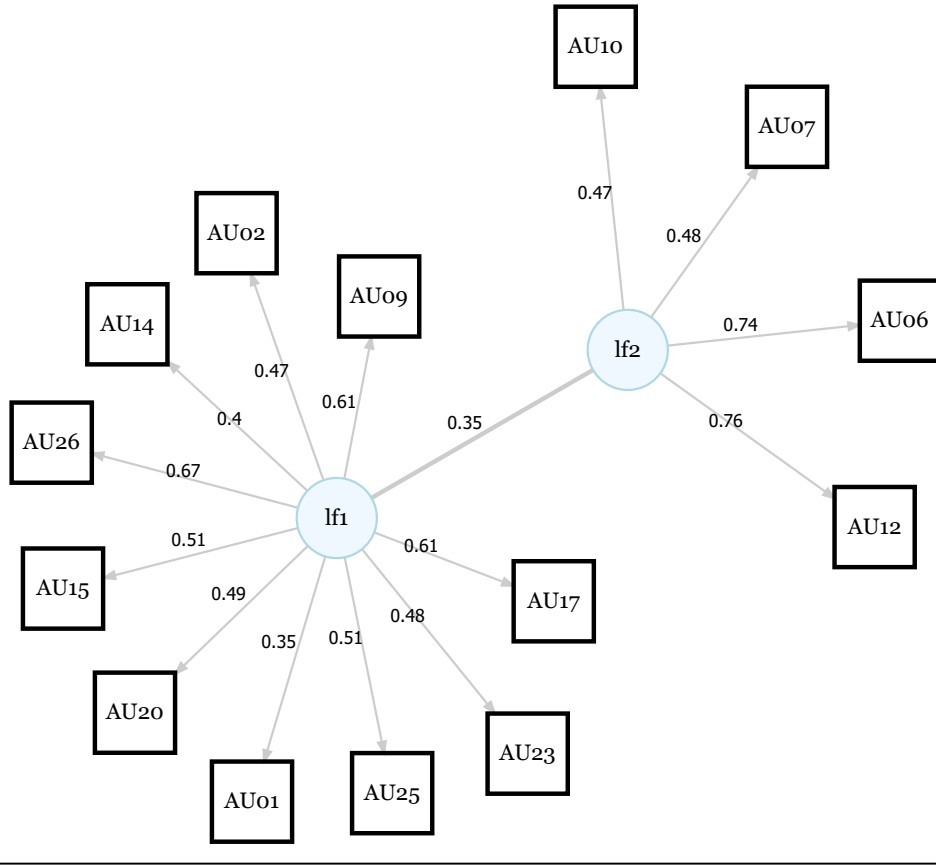

**STAI**. The majority of adolescents peaked right before (53.2%) or immediately after the TSST (37.7%). There was anecdotal evidence for a null-effect of the negative (Est. = 103.11, CI [−10.15, 216.37], $p$ = 0.074, BF = 0.96) and credible evidence for a null-effect of the positive (Est. = −19.71, CI [−132.60, 93.18], $p$ = 0.728, BF = 0.26) latent mimicry scores on adolescent STAI AUCi. Regarding the main effect of parental on adolescent heart rate AUCi, we found credible evidence for a lack of STAI stress-resonance (Est. = 36.86, CI [−67.72, 141.45], $p$ = 0.484, BF = 0.30). Additionally, there was credible evidence for a lack of moderation by the negative (Est. = 5.48, CI [−158.44, 169.41], $p$ = 0.947, BF = 0.17) and credible evidence for a lack of moderation by the positive (Est. = 27.64, CI [−78.52, 133.80], $p$ = 0.605, BF = 0.05) latent mimicry scores.

**Empathic concern**. The majority of adolescents peaked right before (35.1%) or immediately after the TSST (32.5%). There was credible evidence for a null-effect of the negative (Est. = 32.30, CI [−65.34, 129.95], $p$ = 0.511, BF = 0.30) and anecdotal evidence for a null-effect of the positive (Est. = −108.55, CI [−206.19, −10.91], $p$ = 0.030, BF = 0.76) latent mimicry scores with adolescent empathic concern AUCi.

**Personal distress**. Last, the majority of adolescents peaked right before (51.9%) or immediately after the TSST (42.9%). There was anecdotal evidence for a null-effect of the negative (Est. = 96.44, CI [−29.69, 222.57], $p$ = 0.132, BF = 0.56) and credible evidence for a null-effect of the positive (Est. = −45.71, CI [−171.50, 80.08], $p$ = 0.471, BF = 0.25) latent mimicry scores with adolescent personal distress AUCi.

## Discussion

The current study investigated the influence of emotional facial mimicry on stress transmission from parents to their adolescent children. Teenagers between 13 and 16 years of age observed their parents undergo a standardized laboratory stressor. Parents and adolescents simultaneously provided multiple samples of cortisol, heart rate, HRV and subjective stress.

Furthermore, adolescents repeatedly reported on their state empathic concern and personal distress. During stress observation, videos of parents' and adolescents' faces were recorded and subsequently analyzed in terms of action unit (AU) synchronicity. As a possible pathway of how stress is transmitted from one individual to another, we hypothesized that a higher degree of emotional facial mimicry in teenagers would be positively associated with their tendency to show empathic stress while observing their parents' stress experience.

### Mimicry

Activity of a given AU is necessarily accompanied by activity in a multitude of other AUs, because in making specific expressions, movements of facial muscles occur together [e.g. smiling via the cheek raiser (AU06) and the lip corner puller (AU12)[22]]. At the same time, specific AUs act in opposite directions. For example, it is highly unlikely to frown (AU04) while simultaneously raising one's eyebrows (AU02). Facial synchronicity was therefore driven by the synchronicity between parents' and adolescents' expressions, which we attempted to model via a two-factor structural equation model. We found a negative latent factor comprised of AU01, AU02, AU09, AU14, AU15, AU17, AU20, AU23, AU25 and AU26, as well as a positive latent factor including AU06, AU07, AU10 and AU12. Surprisingly, AU04 (corrugator supercilii or frowning muscle) was neither included in the first nor the second latent factor, although corrugator activity has been used as a measure of negative affect in various studies, including studies investigating mimicry[19,23] and facial expressions during psychosocial stress[51]. However, the corrugator has shown weak associations in terms of emotional mimicry[19,23] and might be less reactive than other muscles during social interaction in general[21,52,53]. Hence, the current study produces further evidence that frowning is not as readily reproduced as other facial expressions.

Our mimicry analysis via WCLR and sync interval detection showed that on average, adolescents most likely mimicked their parents' activity at a time lag of about 1.31 s, which matches with previous assumptions of facial

activity being reproduced up to 2 s following the initial behavior[21,23]. However, it needs to be noted that WCLR and peak-picking results heavily rely on the specific WCLR setup used and that countless additional combinations can be tested with greater window sizes, higher maximum lags, and smaller window and lag increments.

## Empathic stress

Overall, evidence for empathic stress transmission from parents to their adolescents was lacking in the current sample, with both cortisol and STAI analyses providing credible evidence for a lack of stress-resonance between parents and their adolescent children. In detail, cortisol trajectories declined over the testing session in adolescents, although parents showed a relatively high proportion of cortisol responders, suggesting successful first-hand stress induction. In adolescents, cortisol responder rates were comparatively low, with only 11.7% exceeding the 1.5 nmol/l threshold. Previous studies have found empathic responder rates of up to 40% in romantic couples[9] and 17.9% in 8–12-year-old children observing their mothers in the TSST[13]. Consequently, the empathic responder rate observed in the current adolescent sample is closer to that of strangers (10%)[9]. Stress is a clearly defined physiological and psychological state that is governed by the autonomous nervous system (ANS) and the HPA-axis. Although secretion of adrenalin or noradrenalin via the ANS is also a necessary condition for stress to ensue, by itself it merely encapsulates a state of arousal[16]. Thus, although an individual is necessarily aroused if it is stressed, it is not stressed because it is aroused. Given the overall decline in cortisol levels, but a time-lagged reaction in autonomic activity and pronounced subjective stress response in the adolescents, we suggest that rather than empathic *stress*, empathic *arousal* was captured.

We suggest that the lack of empathic cortisol responses in adolescents could have been driven by the relationship under investigation. First, cortisol responder rates of adolescents observing their fathers (5.55%) were lower than those of those observing their mothers (17.1%), which were comparable to those of middle-aged children observing their mothers (17.9%)[13]. Although this difference is merely of descriptive nature, maternal stress may be more familiar and relevant to adolescents than paternal stress, because although allocation of primary care-giving has shifted towards a more balanced distribution of fathering and mothering[54], socially constructed roles remain prevalent. Fathers are often viewed as protectors, while mothers continue to be seen as the nurturing parent[55]. This is evidenced by persistent behavioral differences, with fathers spending more working hours and mothers investing more time in child-care[56]. Importantly, this dependence on primary care giving may not be sex—but rather role-specific and needs further research.

A second possible explanation may be the extent of helping behavior that is expected of adolescents. In general, triggering helping behavior may be one of the main functions of empathic stress occurrence[3]. However, since children are typically not responsible for alleviating their parents' stress, stress transmission might be perturbed rather than boosted in this specific relationship. Compared to younger children, it may be even less likely to occur in adolescents striving for independence from their parents[57].

## Empathic stress and mimicry

Adolescent cortisol, heart rate, HRV, STAI, and personal distress were found to be unaffected by the positive mimicry factor. In contrast, associations between the negative mimicry factor and vicarious stress reactivity or stress-resonance revealed largely anecdotal evidence for the null hypothesis.

Mimicry specific to smiling could have been perturbed by increases in empathic stress, given recent evidence of a negative association between acute psychosocial stress and zygomaticus mimicry[52]. In their study, participants' zygomaticus and corrugator mimicry were measured in reaction to videos using EMG at a baseline (outside of stress) and after psychosocial stress induction via the TSST. They found that increasing levels of cortisol were negatively associated with reciprocal smiling, suggesting reductions in zygomaticus mimicry during acute psychosocial stress. The relative lack of cortisol reactivity in the adolescents of this study, however, does not support the notion that their empathic stress contributed to reductions in facial mimicry or vice versa. Rather, adolescents showed little to no significant empathic stress reactions but rather decreases in HRV as well as increases in heart rate and subjective stress experience.

To go on, the transmission of parents' first-hand stress might not have been specific to the stress task itself, rendering the time period in which mimicry was captured too short for the reliable prediction of empathic stress occurrence. This argument is supported by the finding that adolescents peaked in heart rate at the time of reunion with their parents rather than during the TSST itself. Hence, it could be actual interaction between adolescents and parents rather than passive observation that may offer a greater likelihood for emotional mimicry, and in turn, a means to resonate with their parents' stress. However, additional research would be needed to support this notion.

Last, the accuracy of our approach to measuring facial mimicry using video-based AU detection (openFace)[37], and WCLR remains uncertain. The combination of methods used in this study needs to be tested in terms of their validity and reliability in further research. Consequently, it remains unclear to what extent our latent factors accurately reflect facial mimicry between parents and adolescents.

## Limitations

The biggest strength of the current study is the complexity with which both facial behavior, as well as physiological and psychological stress, were recorded. In comparison to studies using electromyography, the usage of digital biomarkers offered the possibility of including 16 distinct AUs, resulting in a more holistic representation of the human face. Furthermore, cortisol, heart rate, HRV, subjective stress as well as state empathy measures were employed to gain a multimodal picture of stress transmission from parents to adolescents. Recruiting fathers next to mothers increased the generalizability of our findings. However, sample diversity was still limited by strict inclusion criteria, especially in terms of health and language requirements and the fact that almost all participants were recruited from one German city.

Unfortunately, data loss in terms of autonomic markers was unexpectedly high. Consequently, power might have been lacking for the heart rate and HRV analyses. Last, although we believe windowed-cross-lagged regression in combination with structural equation modeling was the adequate way to index emotional mimicry in the current data, it is still a complicated and noisy procedure, highly dependent on the initial settings chosen for analysis (e.g. window-size).

## Conclusions

In summary, our findings provided little evidence that emotional mimicry during the TSST was associated with stress transmission from parents to their adolescent children. These null findings could possibly be explained by the overall lack of a full-blown empathic cortisol stress response in adolescents, the decision to measure mimicry only during the TSST rather than when parents and adolescents are reunited following the TSST, and the approach used to calculate emotional mimicry employing video-based AU detection. We conclude that while the usage of digital biomarkers such as AU activity appears to be a promising tool in the study of emotional mimicry, further studies using a similar approach are essential to establish the validity and reliability of the methods employed. Especially the feasibility and cost-efficiency of video recordings compared to bio-physiological measures such as electromyography provide a compelling argument to expand this line of research. Given that "one cannot communicate"[58], non-verbal communication must be recognized as a crucial research avenue, not only in the transmission of psychosocial stress but in human interaction as a whole.

## Data availability

Data is publicly accessible at https://osf.io/qg9yc/.

## Code availability

Analysis scripts are publicly accessible at https://osf.io/qg9yc/.

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

## Acknowledgements
We are thankful to the team of the Social Neuroscience lab at the Institute for Psychosocial Medicine, Psychotherapy, and Psychooncology in Jena, particularly Anke Berger, Ruth Marheineke, Katja Höhne, and most importantly, Hazel Imrie for their support in running the study. We are also thankful to our team of student research assistants, particularly Tina Wenzel, Laura Beuthien, Ann-Christin Winter, Theresa Müller, Ronja Uhlig, Laura Ramirez, Tuba Korkmaz-Walther, Jacob Schuster, Lena Kuschel, and Julia Thaler. This study was funded by a project grant to Veronika Engert (EN 859/3-1) and Philipp Kanske (KA 4412/5-1) from the German Research Foundation (Deutsche Forschungsgemeinschaft). The funders had no role in study design, data collection and analysis, decision to publish or preparation of the manuscript.

## Author contributions
Veronika Engert secured the funding (grant number EN 859/3-1 from the German Research Foundation), designed the experiment, planned the study, executed the study, supported data analysis and supported drafting of the manuscript. Philipp Kanske secured funding (grant number KA 4412/5-1 from the German Research Foundation), designed the experiment and supported drafting of the manuscript. Jost Ulrich Blasberg planned the study, executed the study, analyzed the data and drafted the manuscript. All authors critically revised the manuscript.

## Funding

## Competing interests
The authors declare no competing interests.
