## [Transparent Peer Review file · Communications Psychology]

Little evidence for a role of facial mimicry in the transmission of stress from parents to adolescent children.

Corresponding Author: Mr Jost Blasberg

Version 0:

Decision Letter:

Dear Mr Blasberg,

Thank you for your patience during the peer-review process. Your manuscript titled "Empathic stress in the family: Mimicry boosts subjective and parasympathetic empathic stress in adolescents" has now been seen by 3 reviewers, whose comments are appended below. You will see that they find your work of interest. However, they have raised quite substantial concerns that must be addressed. In light of these comments, we cannot accept the manuscript for publication, but would be interested in considering a revised version that fully addresses these serious concerns.

We hope you will find the Reviewers' comments useful as you decide how to proceed. Should additional work allow you to address these criticisms, we would be happy to look at a substantially revised manuscript. If you choose to take up this option, please highlight all changes in the manuscript text file, and provide a detailed point-by-point reply to the reviewers.

Editorially, we consider it important that you address the concerns raised by Reviewer 2 regarding HF_HRV in the adolescents and the subsequent analyses using this variable. All other methodological and analytic concerns raised by the reviewers should also be addressed. Please add a sensitivity analysis, clarify exclusion criteria and reasoning, and avoid unwarranted causal and directional claims.

Please ensure you follow our statistical guidelines when reporting statistics (<https://www.nature.com/commspsychol/submit/submission-guidelines#statistical-guidelines>). Please note in particular our requirements for the reporting and interpretation of null-results. Non-significant findings derived from null-hypotheses significance tests should be reported in full but may not be interpreted. Where you interpret null results, this interpretation must be based on Bayes Factors or equivalence tests.

I am attaching a checklist that details critical reporting requirements for the revised manuscript. Please attend to each item and ensure your manuscript is fully compliant. We are requesting that your manuscript aligns with these requirements as this facilitates the evaluation of your manuscript, reducing delays in re-review and potential future acceptance. If your revised manuscript is not aligned with these requests on major issues, such as those concerning statistics, it may be returned to you for further revisions without re-review. Additional information can be found in our style and formatting guide Communications Psychology formatting guide.

If the revision process takes significantly longer than five months, we will be happy to reconsider your paper at a later date, provided it still presents a significant contribution to the literature at that stage.

Please use the following link to submit your

- revised manuscript,
- point-by-point response to the referees' comments,
- cover letter (as a separate document),
- the Editorial Policy Checklist (see below),
- the Reporting Summary (see below), and
- the completed Editorial Request Table (attached):

Link Redacted

Thank you for the opportunity to review your work.

Best regards,

Jennifer Bellingtier

Jennifer Bellingtier, PhD
Senior Editor
Communications Psychology

REVIEWER EXPERTISE:

Reviewer #1 interpersonal emotions, cross-lagged models

Reviewer #2 stress physiology, emotions

Reviewer #3 stress, facial expressions

REVIEWER REPORTS:

Reviewer #1 (Remarks to the Author):

Review for

"Empathic stress in the family: Mimicry boosts subjective and parasympathetic empathic stress in adolescents"

This study investigates in 77 dyads of parents and their adolescent children, whether emotional mimicry (here of emotional facial expressions) modulates empathic stress transmission in a standardized stress task (children observing their parents undergoing the TSST). The results are mixed with some limited support for the hypothesis that mimicry plays a (causal) role in the transmission.

This study is overall well written. With some reservations I do see that the study provides an important addition to an emerging field of research. I also appreciate that the authors adhere to several important steps of open science to increase the reproducibility of their work including preregistration and shared data/analysis code. I have several remarks regarding different aspects that are intended to support the strengthening of this paper in a major revision.

Introduction:

(1) In the introduction, the authors could increase the comprehensibility by providing clear definitions of the key concepts. Whereas a definition for empathy is given right at the outset, this definition (seemingly) lacks its cognitive empathy portion. Later on, "cognitive empathy" is referenced without further explanation.

(2) I found the paragraph introducing the mimicry construct and research rather confusing. Reorganizing this, maybe by starting with the Hess & Fischer model, could help against this.

(3) It seems as if the justification for conducting the study in a family context was primarily the high degree of familiarity between the interactants. This is not entirely convincing in my opinion and the authors should make the relevance of investigating this sample clearer in their introduction. One way of doing that could be to strengthen the argument for familiarity by introducing research that has investigated this in relation to mimicry or other interpersonal processes (especially empathy). Another route to go would be to explain, which familial processes may be facilitated by empathic stress abilities. I also assume that the authors are versed in developmental psychology and providing some background on developmental aspects that could play a role here may help justify the approach. Otherwise one might justifiably ask, why not investigate good friends, siblings, co-workers, etc...

Methods:

(4) P. 7, l. 117: Typo? Should this be N=77?

(5) P. 7, ll. 124-128: the authors should add a half-sentence on the reasoning behind the exclusion criteria (which were probably implemented to ensure valid cortisol measurement?)

(6) P. 8, l. 146: Typo? Should this be 7.5% before the onset of their menarche?

(7) P.9, ll. 168-170: Could the authors explain the passive observer role of the adolescents a bit further? What does it mean/how was it described to them that they should abstain from nonverbal communication? Obviously, this could also make them act unnaturally with regard to facial expressions. Suppressing emotional states, which may be necessary also, could also induce stress, other than the empathic stress aimed for in this research. I also wonder, whether parents were aware of the instructions given to their children. In the spirit of the TSST - probably not. Can the authors clarify?

Apart from this, "they were not allowed to.." may be better phrased as "they were instructed to not.." – at least I would hope this was true.

(8) P. 14, ll. 282-283; 282; "Because in our study adolescents were observing their parents, synchronization could only occur in one direction.": Given that the adolescents were visible to their parents, this is not correct. The researchers, may, however only be interested in one direction of mimicry, and could justify that. However, it is very likely that mimicry also occurred in the other direction. In fact, the authors could easily check whether there was a difference between pacing and leading by also calculating parents lagging behind their children.

(9) P. 17, l. 339: I assume that the two factor solution was used based on the scree-plot. [Later] I have found this information in the results. Would be great if the authors stated this explicitly somewhere between ll. 338-339 to ease comprehension.

(10) For the regression models, were continuous predictors centered to their mean when including them in interaction terms? This would be indicated and should be corrected if this was not the case.

(11) P. 18, ll. 361-362: Can the authors give any insight into the reasons behind the high number of invalid HR recordings in adolescents? A good general rule to deal with missing values is to try to prevent them from occurring and there may be reasons why this failed in this instance?

Results:

(12) P. 19, Figure 3: Please adjust the x-scale of the HR/HRV data to somewhat represent the timing of the study. This would help interpretation in context with the other measures. If this is not feasible, it could help if there were indicators for the start and end of the TSST in each subplot.

(13) P. 20; WCLC: A common test for WCLC synchrony is, to check for statistical significance of the synchrony scores against some form of pseudo-synchrony. This check makes it able to distinguish interpersonal synchrony from synchrony that occurs by chance and in the case of this study also synchrony that occurs as a result of mere procedural aspects of the study. Examples for such pseudo-synchrony checks can be found in the extant literature (Ramseyer & Tschacher, 2011; Altmann, 2013; Riehle et al., 2017). A fairly easy solution could be to check for synchrony in video recordings of adult/adolescent combinations that are not parent and child, i.e. who did not actually interact with one another. Relatedly, Altmann (2013) used the problem that two participants respond to a similar stimulus and are in interaction with one another as rationale for his windowed cross-lagged regression. I am unsure if the interpersonal situation in the present study poses this exact problem, as the adolescents are not in the TSST situation themselves. Still, I would appreciate if the authors dealt with aspects of synchrony that could occur for reasons other than empathic responding/mimicry somehow.

(14) P. 20, WCLC: Apart from the correlations, it would be great if the authors could also present M (SD) for each AU - this is an important qualifier for synchrony interpretation. The results suggest, for example, that AU20 was synchronized for about 25% of the time, but this AU was probably not shown for 25% of the time. In fact, it is even good practice to control for both interactants' mean values when testing for effects of synchrony. This could be difficult in the present approach using the EFA, but one could try and extract latent factor scores for AUs' means based on the AU allocation to the two latent mimicry factors. At least presenting Ms and SDs in the supplement and interpreting this would be great for having that context.

(15) P. 21, ll. 405-406: In my opinion, the negative expression mimicry factor should be labelled only "synchronization with negative expressions". That it is comprised of various negative emotions does not necessarily mean that a synchronization with a complex expression has actually occurred.

(16) P. 22, l. 416 and onwards: For all models that include a main effect of parental AUCi on adolescent AUCi, please also include the results for these main effects in the text and include these in the interpretation where adequate.

(17) P. 23, ll. 424-425; "The majority of adolescents peaked immediately after (40.3 %), 10 minutes after (23.4 %) or 20 minutes after the TSST (13.0 %)": Figure 3 suggests a steady decline in Cortisol in adolescents over time - on average... how does this fit with these results? Of course individual trajectories maybe cancelling out one another - but not if some 75% peaked in cortisol after the TSST. Would it not be fair to say that adolescents did not really show a Cortisol response?

(18) P. 25, Fig. 6: The authors interpret this interaction as counter to their hypothesis, but is this not in line with the hypothesis, if mimicry was low and therefore did not facilitate empathy and adolescents then did not respond with low HRV (high stress) themselves?

Discussion:

(19) P. 26, ll. 492-493; "one cannot frown (AU04) while at the same time raising one's eyebrows (AU02)": I am sure that some people can and would advise against such absolute statements in favor of more probabilistic ones.

(20) P. 26, l. 498; "corrugator fascilii": This one is more commonly known as Corrugator supercilii.

(21) P. 27, ll. 508-510, sentence on facial feedback hypothesis: I disagree with this statement, because it insinuates that the findings were evidence of a causal relationship between mimicry and affective-physiological state. The data do not prove the temporal relationship of this putative causal direction (i.e., it is not clear if mimicry/emotional expression caused the internal affective state or if the affective state caused emotional expression) and the data also do not show that there is a causal link, even if we assumed this temporal correlation to be in line with the putative causal direction. It could be, for instance, that shared knowledge about emotions and mentalization (i.e. cognitive portions of empathy) cause internal affective states in both the parent and the adolescent, which both of them then express similarly on their faces based on shared display rules. Now - I do consider some of the findings in line with the hypothesis of a facial feedback or mimicry being a potential causal factor driving this. However, before one may call this causal, more (experimental) research needs to follow. As this interpretation is appearing in several ways in the discussion, I would advise to revise the discussion and tone down these claims of causal directionality and causality.

The concluding sentence in the abstract should also be revised accordingly.

(22) P. 28, ll. 537-539; "Actual interaction between adolescents and parents rather than passive observation may thus offer the possibility for bi-directional emotional mimicry, thus boosting empathic stress occurrence.": Again, I would argue that back-communication occurred on some scale - otherwise the authors would not have been able to detect facial expressions in the adolescents.

Writing this comment brings me to another point: Is there any way to quantify the reliability/internal consistency of the OpenFace AU detection?

(23) P. 29, l. 547-548; "Recruiting fathers next to mothers increased generalizability of findings to households with

heterosexual parents.”: I do not understand what this has to do with the question of whether or not the household includes hetero- or homosexual parents? The sampling increases the generalizability from only mothers to also fathers, which is a noteworthy strength, but this is irrespective of sexuality.

Conclusion:

(24) P. 29, ll. 560-561: Here, conclusions are drawn based on the whole set of hypotheses but based on some singular positive in light of several negative findings. This would constitute a case where FWE-correction is indicated even according to García-Pérez (2023).

(25) P. 30, l. 568; “shared on a sub-conscious perceptual level”: I do not think that this study has necessarily tested mimicry as a sub-conscious construct and, as mentioned above, one could even argue the necessity of a perceptual aspect of what has been assessed (i.e., anticipation processes based on generalized expectations play into the expression). Maybe this could be stated more neutrally.

Reviewer #2 (Remarks to the Author):

The manuscript outlines a study that was designed to measure facial mimicry between parent and adolescent while the parent was being stressed. The goal of this study was to elucidate a mechanism by which stress can be transmitted from one individual to the next. The authors of this study recruited 76 dyads to the laboratory. Cortisol, heart rate, HF-HRV and a number of subjective stress measures were used in addition to being videotaped to assess facial mimicry.

First, I would like to point out how impressed I am with the methods of this study. Synchrony work can be extremely difficult and often end up in the weeds, so to speak, but as I read through this manuscript I was thoroughly impressed and satisfied with all methodological choices made by the authors. So on this note, job well done to the authors who made well-informed and discerning choices throughout the analyses.

My main critique of this work is twofold:

1) HF-HRV plays a centrepiece role in this paper – yet its calculation was likely done improperly. HF-HRV in adolescents must be calculated using a different frequency spectrum due to the heightened breathing rate of adolescents, as compared to adults. For example, a typical adult breathes 9 breaths/min while a 12-year old breathes at 14 breaths/minute and a 16-year old at 11 breaths per minute. Within the methods section, the authors did not specify the frequency window used when assessing HF-HRV (usually is between 0.15 and 0.40 Hz for adults). Firstly, this needs to be specified. Secondly, due to the heightened breathing rate of adolescents, this frequency window needs to be modified for the adolescent participants based on their age. So for example HF-HRV in a 12 year old wouldn't be >0.15Hz, it would be >0.23Hz instead. These analyses need to be redone accordingly.

Alternatively, the authors can shift their analysis to using RMSSD. This is not a frequency-based measure of HRV, and thus is independent of breathing rate. RMSSD would not need to be corrected for due to age, therefore yielding it as a potentially superior measure of HRV in this specific context, where HRV is needed to be compared across generations.

I am also a bit perturbed by the massive drop in n due to issues with the cardiac data. Dropping 34 dyads (out of 76) is dropping 45% of the data. I would like to see more than one line dedicated to this issue. Understood sometimes data is messy and cannot be used, or there are technical difficulties, but 45% is outside of the norm, and at least an extra line or two explaining this would be helpful.

2) My second main qualm is the title and conclusion of this paper which states that mimicry boosts empathic stress responses. While this paper has very interesting findings (I was enthralled!) – I did not see this as a takeaway from the paper. It certainly would have made for a neat picture, but facial mimicry did not seem to be related to any of the subjective stress measures except for the STAI, and this was in line with the vicarious stress model, not the stress resonance model. So my takeaway from this is that adolescents who experienced more anxiety watching their parents get stressed, had more facial synchrony with them (for negative expressions). The directionality of this isn't necessarily obvious – couldn't it be that adolescents who are more emotionally reactive (scoring high on STAI while watching a stressed parent) experience more facial mimicry? Facial mimicry was related to HF-HRV (results tbd – maybe they will become more intuitive with age correction) but as of now, the results yield an intriguing relation, but not one that supports the claim that empathic stress has been boosted. If anything, I believe the interesting takeaway from this paper is that facial mimicry isn't more related to each of the measures (cortisol synchrony, heart rate synchrony etc.) That being said, the authors chose to use AUC measures for heart rate and HF-HRV – synchrony of these measures could be done in a more fine-scaled way. I can understand why that would overload this paper – its analyses are already very hefty, but it would certainly be interesting! Basically, my feedback is to tone down the conclusions of the paper and keep them more in tune with the actual results. I don't think that takes away from the clearly impressive package that this paper is.

Minor issues:

1) In the abstract it says n = 77, and in the manuscript it says n = 76, which is it?

2) Definition of empathy presented in first line is more so a definition of affective empathy than an all-encompassing definition of the phenomenon.

3) Line 65 ends without a citation, and it is unclear if they are referencing Miller et al. 2013 or Engert et al., 2014

4) The concept of “action units” was not talked about enough in the discussion. In line 103 when it is mentioned it is unclear what the authors are talking about. I would find a space earlier in the intro to write a line or two about action units and what they are.

5) I would refine the last line of the introduction (“Based on the role of mimicry..”) it's unclear and can be better written

- 6) Dyads rested for 40 minutes. What did they do in that time? Were they allowed to talk? Did they have reading materials?
- 7) Line 194, when the TSST is introduced, be sure to indicate that it's been shown to elicit a robust stress response
- 8) Line 235; I wouldn't mind seeing one or two examples of the questions or adjectives used in the ERS, for those unfamiliar with it
- 9) Why isn't AU 25 included in Fig. 2?
- 10) I like how level of synchronization was operationalized by time in sync...but what about the actual level of the correlation? Why was that not looked at at all?
- 11) Why were sync intervals shorter than 1 second excluded?
- 12) Figure 3 has inconsistent x axes – can they all be in minutes? That would be more intuitive
- 13) I can safely assume that the 34 missing dyads only affected the HF-HRV and HR data and all dyads were included in all other analyses, right?
- 14) Typo Line 491 (lip corner not lip corner)
- 15) Line 561 “synchronizing with their caregiver's facial activity increased their empathic stress reactivity” – did it? From my reading I didn't see this, and this harkens back to the second main issue I brought up above

Kudos:

- Using audio waveform to synchronize the videos is very smart! Nice work
- window sizes, lags and increments were all nicely chosen
- Bonforonni at $p < 0.0000006$ is great. From my calculations may even be an overcorrection? But nice work being conservative
- Loved the idea of using an EFA to derive to latent factors from the Aus
- Love the inclusion of Fig. 3 to see the dynamics of all these measures across dyads – many papers seem to omit such basic graphs but they really add to the manuscript
- Interesting to see the finding that on average 24% of the time was spent with facial mimicry! Just this finding alone is laudable given the extensive analyses needed to arrive at this number
- well-written and clear
- great choice of statistical methods and excellent word clearly spelling them out

Overall I was especially impressed with this paper but my two main issues are that the HF-HRV data must be age-corrected or changed to RMSSD and the discussion/title abstract should be edited to better represent the actual findings of the paper (which may change with the HF-HRV analyses). I think this paper can be influential to the field in showing a very well-laid out method of analyzing synchronous facial muscle activity from video footage; this opens the door to many interesting avenues in the empathy domain.

Reviewer #3 (Remarks to the Author):

The current paper is concerned with the spontaneous effect of stress spillover, specifically, the role of mimicry in the transmission of stress. The authors examined a sample of adolescents who observed their parents undergo a stressful situation. Looking at young adults is particularly interesting as it fills an important gap in the literature. Similarly, looking at mimicry is an interesting way to quantify contagion, and in particular contagion in familiar dyads. Overall, this study is a timely and important contribution to the literature that will help us to better understand the link between stress and close relationships.

I first want to congratulate the authors for an interesting and engaging manuscript. The methodological aspect of the studies is expertly done and in line with the high standards for stress research. As such, the manuscript has many strengths and could make an important contribution to the field. The main drawback is the small sample size for the autonomic markers—the ones that are statistically significant. This issue must be addressed more clearly and even added explicitly to the abstract. In addition, I would like a more thorough discussion of the relatively small stress response in the observers. This seems to deviate from previous studies (also from the same group using a similar design). This is not a problem per se, but it is curious/interesting enough that it should warrant some attention. In my opinion, the results (and rigorous method) still warrant a possible publication, despite this. There are some additional issues that I want to highlight to improve the paper before I can recommend it for publication. Together with some more general comments I have added them below.

Comments:

There is a recent review (Nitschke & Bartz, 2023; Neuroscience and Biobehavioral Reviews) that has reviewed the evidence of empathic stress (or stress contagion) that goes a bit beyond the (admittedly excellent) summary by Engert et al. (2019) by discussing the importance of familiarity (or closeness) for the spillover effect in more detail.

I greatly appreciate the transparent preregistration and the open data. This is excellent and should be acknowledged as such.

Sample size. It is unclear how many participants were excluded due to various issues mentioned in the manuscript. There should be a table in the Supplemental Materials indicating the exact participant/dyad number for each DV.

There also seems to be a mismatch between parents ($n=76$) and children ($n=77$). In addition, the percentage reporting for mothers is confusing and would benefit from additional information (i.e., 15.8 % of all parents; 9.5% of mothers). I also don't know what phase of the menstrual cycle the 28.5% of mothers were in that was not reported (same with daughters—the numbers do not add up).

I am not sure that providing all participants with orange juice and chocolate will equalize their blood sugar levels—specifically, also because this “equalization” was not tested.. In my opinion, this approach introduces more confounds than a spontaneous, unaltered HPA axis response. Yes, a manipulated HPA axis response might look nicer on paper, but there is insufficient evidence to attribute this spike in cortisol solely to a higher metabolic potential. Of this is to say, this approach makes studies less comparable and glucocorticoid-specific effects less confident. I do not know if this approach standardizes metabolic output potential. That said, I appreciate the disclosure, and this transparency is sufficient to address methodological differences in stress induction, particularly in the glucocorticoid stress response between studies. I do not need this to be addressed any further.

All other aspects of the stress protocol of the study are expertly done.

In regards to the handling of the cortisol data. The skewness of raw data is not very important for linear models (or not important at all) as it is rather about the skewness of the residuals from the model. As such, it is not necessary to preemptively transform the raw data by applying a log transformation. Ideally, the authors should check if including raw cortisol data in the models violates the homoscedasticity assumption. Similarly, it might be a good idea to run the models without winzoration first. In my opinion, stress manipulations will naturally produce extreme values—and most of these extreme values will carry important information that should not be preemptively dismissed (or reduced). I do not need the authors to change any of their analyses, but, at the very least, the authors should provide numbers on how many data points were altered.

Apart from the previous comment, I do not have any issues with the statistical approaches. They appear to be expertly executed.

I understand the rationale for not correcting for multiple comparisons. Still, I find it a bit odd as a blanket statement since some dependent variables clearly measure similar things—e.g., in the case of, autonomic activation, there are two measures: HR and HRV). Otherwise, I tend to agree.

The authors should include how many data points (percentage) were imputed. It is also important to include the exact method (including packages if conducted in R).

The power analysis needs further clarification. The (updated) pre-registered power analysis seems very reasonable and estimates that “a minimum sample size of $N = 64$ dyads is required” to find a moderate effect ($f^2 = 0.1$). This is more or less in line with the power estimation here. However, noticeably, the sample reported on ($n=76$; not sure how many/ or any were excluded) is smaller. I understand that the authors accounted for drop-outs in their OSF pre-registration, but this is not very well reported in the manuscript. The authors should clarify this and also make a short assessment of the current sample regarding the power analysis.

It would be helpful to include the time of observation in the figures (including the durations), as is, this is not very informative, especially given the different x-axis, and would benefit from more information.

For the Supplemental Materials, the table would benefit from additional information (i.e., notes at the bottom) to explain the variables.

Overall I like the points from the discussion and the interpretation of the findings in the context of other research. However, the discussion is a bit disjointed and would benefit from a thorough readthrough. All the ideas are there, but they do not necessarily build on each other. I also wish that some of the ideas (or cited research) were described in more detail, as it would help to put the research in context. I am also missing a big-picture assessment of the findings. For a more general journal, such as *CommPsych*, this is necessary as not every reader will be as in the weeds of stress research as the authors (or the reviewers), it would help make the findings more accessible (this could also benefit the intro, to some extent).

I would focus more on the fact that empathic stress was not very high in this adolescent sample. This may be due to the sample, the nature of the age of participants, and some methodological choices. It might also be important to put this finding in context, notably, how many adolescents showed a stress response (or empathic response)? Can this be quantified similarly to previous papers on this topic (e.g., Engert et al.)? For example, for Cortisol, studies on stress contagion have shown increases in 7%-40% of the observer sample (see Nitschke & Bartz, 2023). For ANS activation, neither Buchanan nor Engert has found meaningful co-activation, but it appears that there (but small and de-synchronized) increase in HR in the current sample of adolescents. The fact that subjective stress levels are strongly co-activated in both observers and targets should also not be dismissed. It further highlights the complexity of the stress response.

It has also been reported that the corrugator (and maybe frowning more generally) might be less reactive in social contexts (Hess & Fischer, 2013; Nitschke et al., 2019; Niedenthal et al., 2010; Carr, 2014), and it might therefore be difficult to pick up on more subtle changes.

In that regard, the reported effect by Nitschke et al., (2020) was specific to zygomaticus activation—stress reduced reciprocal smiling, while it did not affect corrugator activation. Importantly, here the observers were actively stressed, whereas the target was unstressed. As such, the data might be especially informative for adolescents who were particularly stressed by observing their parents (i.e., AUCi-adolescent—mimicry-smiles).

EDITORIAL POLICIES

We ask that you ensure your manuscript complies with our editorial policies and reporting requirements.

To that end, we require revised manuscripts to be accompanied by two completed items: a reporting summary that collects information on study design and procedure, and an editorial policy checklist that verifies compliance with all required editorial policies

- <https://www.nature.com/documents/nr-reporting-summary.zip>>Nature Research Reporting Summary
- <https://www.nature.com/documents/nr-editorial-policy-checklist.pdf>>Editorial Policy Checklist

All points on the policy checklist must be addressed. Your revised manuscript can only be sent back to the referees if these checklists are completed and uploaded with the revision.

Notes: If you have submitted a Stage 1 Registered Report, Review, Primer, Comment, or Perspective you do not need to submit these forms. If you have already submitted these forms, you may disregard this request.

** Visit Nature Research's author and referees' website at <http://www.nature.com/authors>>www.nature.com/authors for information about policies, services and author benefits**

If you experience problems in linking your ORCID, please contact the <http://platformsupport.nature.com/>>Platform Support Helpdesk.

Version 1:

Decision Letter:

Dear Mr Blasberg,

Thank you for your patience during the peer-review process. Your manuscript titled "Empathic stress in the family: Investigating facial mimicry in the transmission of stress from parents to adolescent children" has now been seen by 3 reviewers, and I include their comments at the end of this message. They are largely supportive of your revised manuscript. However, upon editorial review, we find that further work is needed before we can make a final decision on the manuscript.

Editorially, we ask that you address the remaining concerns of Reviewer 1 and provide additional evidence and documentation in line with our journal policies.

For manuscripts that interpret null results, we require Bayes Factors or equivalence tests to interpret the null results—these

must be added. Furthermore, we ask that you add a sensitivity analysis to your manuscript. Please do not conduct a post-hoc power analysis based on the observed effect size in your study (cf. Lakens, 2022, <https://doi.org/10.1525/collabra.33267>). Our full guidelines on statistical reporting are available here: <https://www.nature.com/commpsychol/submit/submission-guidelines#statistical-guidelines>

It is our policy that authors must disclose all deviations from the preregistered protocol and explain the rationale for deviation (e.g., flaw, feasibility, suboptimality). In cases of deviation from the preregistered analysis plan for reasons other than fundamental flaw or feasibility, the originally planned analyses must also be reported. Please report the results from the preregistered WCLC models in the Supplement. We ask you to state that the correction for multiple comparisons was not preregistered. You may add that this was undertaken in response to a reviewer request. Additionally, we ask that you highlight the deviation from the preregistration regarding the analysis-informed application of log transformations for skewed variables and the absence of winzorising. You may state that this change was implemented in response to a reviewer suggestion. You can find our full policy on preregistration here: <https://www.nature.com/commpsychol/submit/preregistration>

We therefore invite you to revise and resubmit your manuscript, along with a point-by-point response to the reviewers. Please highlight all changes in the manuscript text file.

Please submit the following items:

- Revised manuscript
- Point-by-point response to the referees' comments
- Cover letter (as a separate document)
- [Nature Research Reporting Summary](https://www.nature.com/documents/nr-reporting-summary.zip)
- [Editorial Policy Checklist](https://www.nature.com/documents/nr-editorial-policy-checklist.pdf)

via this link: Link Redacted .

Additional guidance is available in our style and formatting guide [Communications Psychology formatting guide](https://www.nature.com/documents/commpsychol-style-formatting-guide-accept.pdf).

Best regards,

Jennifer Bellingtier

Jennifer Bellingtier, PhD
Senior Editor
Communications Psychology

REVIEWER EXPERTISE:

Reviewer #1 interpersonal emotions, cross-lagged models
Reviewer #2 stress physiology, emotions
Reviewer #3 stress, facial expressions

REVIEWER REPORTS:

Reviewer #1 (Remarks to the Author):

I want to congratulate the authors on such a thoughtful and constructive revision of their manuscript, which has now significantly gained in strength. I do have three comments left pertaining to the WCLR analysis. These may not warrant an entire revision cycle, however:

p. 18: Typo in "We chose similar parameters as Riehle et al. (2017), the currently one other study investigating facial mimicry using WCLR.": We actually used WCLC, not ...R in this paper.

p. 18: In "Subsequently, all negative slopes and slopes with an $RCC^2 < 1$ were dropped from the RCC^2 matrices.": Is this value (< 1) correctly stated? R^2 would not increase beyond 1, or am I missing something?

WCLR analysis: The authors now have used an absolute R^2 threshold of 0.25 for their synchrony measure based on Schoenherr et al. (2019). It could be confusing that the authors also state that the significance of synchrony is determined based on the significance of the delta R^2 and then use an effect size cut-off. Including a qualifying sentence could help here. It could explain that this cut-off is valid for the window size used in the analyses (which it actually is, considering power aspects and correction for 756075 parameter tests).

Reviewer #2 (Remarks to the Author):

I would like to commend the authors for doing an excellent job at implementing the changes all the reviewers suggested. I'm a big fan of the paper in its current state, job well done! The work done to create this paper is enormous, and I believe it lays a great groundwork for any other future papers that want to analyze mimicry in this way. The results weren't significant, but I think that's interesting in its own right - more mystery about how stress contagion occurs! - and I think your paper sets a good example of reporting important work in a high impact journal, even if those results aren't significant. In my view this paper is very significant and I am looking forward to see this published.

Reviewer #3 (Remarks to the Author):

I am content with the author's responses to my concerns and appreciate the inclusion of some of my suggestions. I have no further concerns and would recommend a publication.

Communications Psychology is committed to improving transparency in authorship. As part of our efforts in this direction, we are now requesting that all authors identified as 'corresponding author' create and link their Open Researcher and Contributor Identifier (ORCID) with their account on the Manuscript Tracking System prior to acceptance. ORCID helps the scientific community achieve unambiguous attribution of all scholarly contributions. You can create and link your ORCID from the home page of the Manuscript Tracking System by clicking on 'Modify my Springer Nature account' and following the instructions in the link below. Please also inform all co-authors that they can add their ORCIDs to their accounts and that they must do so prior to acceptance.
<https://www.springernature.com/gp/researchers/orcid/orcid-for-nature-research>

Version 2:

Decision Letter:

Dear Mr Blasberg,

Your manuscript titled "Empathic stress in the family: Investigating facial mimicry in the transmission of stress from parents to adolescent children" has now been editorially reviewed, and I am delighted to say that we are happy, in principle, to publish a suitably revised version in *Communications Psychology*.

We therefore invite you to revise your paper one last time to address a list of editorial requests. At the same time we ask that you edit your manuscript to comply with our format requirements and to maximise the accessibility and therefore the impact of your work.

EDITORIAL REQUESTS:

SUBMISSION INFORMATION:

OPEN ACCESS:

*** TRANSPARENT PEER REVIEW:** *Communications Psychology* uses a transparent peer review system. On author request, confidential information and data can be removed from the published reviewer reports and rebuttal letters prior to publication. If you are concerned about the release of confidential data, please let us know specifically what information you would like to have removed. Please note that we cannot incorporate redactions for any other reasons.

*** CODE AVAILABILITY:** All *Communications Psychology* manuscripts must include a section titled "Code Availability" at the end of the methods section. We require that the custom analysis code supporting your conclusions is made available in a publicly accessible repository at this stage; please choose a repository that generates a digital object identifier (DOI) for the code; the link to the repository and the DOI must be included in the Code Availability statement. Publication as Supplementary Information will not suffice.

*** DATA AVAILABILITY:**

All *Communications Psychology* manuscripts must include a section titled "Data Availability" at the end of the Methods section. More information on this policy, is available in the Editorial Requests Table and at <http://www.nature.com/authors/policies/data/data-availability-statements-data-citations.pdf>.

Link Redacted

Best regards,

Jennifer Bellingtier

Jennifer Bellingtier, PhD
Senior Editor
Communications Psychology

Dear editor and reviewers,

First, we would like to thank you for your constructive and critical comments on our initial submission.

Not only has the quality of our manuscript greatly improved due to your helpful suggestions, but various concerns lead to us finding some errors in our initial data analysis. Thus, we are very very thankful to the reviewers for pointing us to this.

To give a quick overview, the following major changes have been made to our methodology and analysis:

1. HRV is now indexed by RMSSD instead of HF-HRV as suggested by Reviewer #2. By doing so, differences in breathing rate across puberty and in comparison to adults can be circumvented.
2. As suggested by Reviewer #1 we have changed our synchronicity calculation from windowed-cross-lagged-correlation (WCLC) to windowed-cross-lagged-regression (WCLR) to control for possible auto-correlation and pseudo-synchrony. We have updated our methods section accordingly.
3. We have changed our multiple testing approach. We are now correcting for physiological (cortisol, HR, HRV) and subjective (STAI, empathic concern, personal distress) stress markers, respectively. Thus, the significance threshold has been set to $p = 0.0167$. Given that physiological and subjective stress measures have been found to be rather disconnected, we decided to interpret them as specific rather than general cases of empathic stress.

As a result, our initial findings concerning effects of mimicry on subjective stress and HRV are no longer present. We would like to note, however, that the overall structure of our mimicry analysis has remained the same and has lead to highly similar results as in our initial submission. In detail, the exploratory factor analysis still lead to a two-factor model with a mimicked positive and a mimicked negative latent factor, and largely the same distribution of observed variables.

Please find detailed answers to your concerns below.

Reviewer #1 (Remarks to the Author):

Review for

“Empathic stress in the family: Mimicry boosts subjective and parasympathetic empathic stress in adolescents”

This study investigates in 77 dyads of parents and their adolescent children, whether emotional mimicry (here of emotional facial expressions) modulates empathic stress transmission in a standardized stress task (children observing their parents undergoing the TSST). The results are mixed with some limited support for the hypothesis that mimicry plays a (causal) role in the transmission.

This study is overall well written. With some reservations I do see that the study provides an important addition to an emerging field of research. I also appreciate that the authors adhere

to several important steps of open science to increase the reproducibility of their work including preregistration and shared data/analysis code. I have several remarks regarding different aspects that are intended to support the strengthening of this paper in a major revision.

Introduction:

(1) In the introduction, the authors could increase the comprehensibility by providing clear definitions of the key concepts. Whereas a definition for empathy is given right at the outset, this definition (seemingly) lacks its cognitive empathy portion. Later on, "cognitive empathy" is referenced without further explanation.

Thank you for pointing this out! We have added brief definitions of affective and cognitive empathy in the introduction to facilitate better understanding of our key concepts.

Page 3: "Beyond the sharing of subjective experience, empathy entails the reproduction of others' physiological and neural states. This can occur through affective empathy, which is the mere reproduction of another's feeling, or cognitive empathy, which combines affect sharing with mentalizing about another's thoughts and feelings (Batson, 2009)."

(2) I found the paragraph introducing the mimicry construct and research rather confusing. Reorganizing this, maybe by starting with the Hess & Fischer model, could help against this.

We absolutely agree that our paragraph on mimicry is quite complex and dense in the amount of information it tries to convey. Accordingly, we have restructured this paragraph, introducing the model by Hess & Fischer in the beginning and hope that it is easier to understand now.

Page 5: "Although the phenomenon of empathic stress has been successfully replicated in multiple studies, we know very little about the intricacies of *how* stress is shared on a behavioral level. Mimicry, the spontaneous reproduction of changes in facial expressions, has been suggested as a subliminal precursor of emotional contagion and empathy (Hatfield et al., 1994; Hawk et al., 2011; Lamm et al., 2008; Murata et al., 2016), and may therefore play a role in the transmission of stress. The emotional mimicry in context model (Hess & Fischer, 2013) theorizes that rather than constituting context-free expression reproduction, mimicry provides functions of social regulation. Importantly, a positive affiliation between observer and target is suggested to be a necessary requirement for emotional mimicry to occur, especially if the mimicked expression is of negative valence (Hess & Fischer, 2013). To illustrate, the context-free reproduction of a nemesis's frown would be counter-intuitive, given that their demise would induce satisfaction rather than sadness. As such, emotional mimicry necessitates a positive affiliation or the explicit motivation to infer another's affective state (Murata et al., 2016).

Investigating mimicry in empathy-inducing situations, Lamm and colleagues (2008) instructed participants to watch video clips of patients receiving painful sonar treatment while either imagining the patients' feelings or receiving the treatment themselves. Corrugator supercilii (frowning muscle) activity measured via electromyography (EMG) increased irrespective of the taken perspective, while activity of the orbicularis oculi (lid tightening muscle) only rose if participants imagined themselves in the same situation. Murata et al.

(2016) corroborated these findings by revealing a higher tendency to mimic facial expressions when participants were directly asked to infer another's affective state, both in terms of activity in four different muscle sites and regarding distinct facial expressions rated using the Facial Action Coding System (FACS; Ekman & Friesen, 1978). The FACS constitutes a taxonomy to categorize movements of facial muscles or muscle groups in terms of action units (AUs).

In one of the few studies investigating emotional mimicry in a real-life dyadic setting, Riehle and colleagues (2017) measured EMG activity of the corrugator supercilii and the zygomaticus major (smiling muscle) in a dyadic discussion task. Using windowed-cross-lagged-correlation (Boker et al., 2002), the authors detected significant covariation only for zygomaticus activity. These findings inform research in the context of psychosocial stress. Both, emotional mimicry occurring when perspective taking is manipulated (Lamm et al., 2008; Murata et al., 2016) and a lack of synchronization for negatively valenced facial behavior between strangers (Riehle et al., 2017), are in line with previous findings regarding lower stress transmission between strangers in comparison to romantic couples (Engert et al., 2014) and higher trait cognitive empathy associating with greater empathic stress occurrence in observers (Blasberg et al., 2023; Engert et al., 2014).”

(3) It seems as if the justification for conducting the study in a family context was primarily the high degree of familiarity between the interactants. This is not entirely convincing in my opinion and the authors should make the relevance of investigating this sample clearer in their introduction. One way of doing that could be to strengthen the argument for familiarity by introducing research that has investigated this in relation to mimicry or other interpersonal processes (especially empathy). Another route to go would be to explain, which familial processes may be facilitated by empathic stress abilities. I also assume that the authors are versed in developmental psychology and providing some background on developmental aspects that could play a role here may help justify the approach. Otherwise one might justifiably ask, why not investigate good friends, siblings, co-workers, etc...

Thank you for this suggestion! First, we would like to very much agree with your sentiment that any of the dyads you mentioned would have a justification to be studied both in terms of psychosocial stress and mimicry, given that good friends, siblings, co-workers and parents are all important interaction partners in everyday life. What distinguishes the parent-child dyad in light of psychosocial stress transmission is that, as you already mentioned, children are still developing, both in terms of their social capacities and their physiological and psychological stress systems. Consequently, empathic stress could both foster adaptive coping and social behaviour in the light of adversity or lead to hazardous life circumstances if the existing stressors cannot be jointly alleviated. Furthermore, the parent-child dyad, like other familiar dyads, offers a (usually) positively affiliated dyad, which is necessary to study emotional mimicry according to Hess & Fischer.

Following your suggestion, we have elaborated on this notion in our introduction with a special focus on developmental psychology:

Page 4: “Due to its high relevance for the development of the human stress system across childhood and puberty (Thompson, 2014), the mother-child bond is of particular interest in the study of stress-resonance (Blasberg et al., 2023; Waters et al., 2017; Waters et al., 2014).

In this context, the pubertal stress recalibration hypothesis (Engel & Gunnar, 2020) states that adolescence serves as a sensitive period, providing both a chance to rejuvenate or worsen the effects of early life adversity. Thus, in an ideal familial scenario, empathic stress provides the necessary energy to motivate appropriate helping behavior and jointly alleviate a stressor at hand (Engert et al., 2019), providing opportunities to safely experience adversity together. If, however, existing stressors cannot be alleviated and parental chronic stress is shared across longer periods of time, stress in the family system might rather spread than dissipate. This would put children at risk of hazardous living circumstances and the early development of stress-associated disease (Thompson, 2014).”

Methods:

(4) P. 7, l. 117: Typo? Should this be N=77?

Fixed.

(5) P. 7, ll. 124-128: the authors should add a half-sentence on the reasoning behind the exclusion criteria (which were probably implemented to ensure valid cortisol measurement?)

Absolutely; we have added the following sentence to clarify the reasoning behind our exclusion criteria:

P. 8: “These criteria were chosen to ensure that cortisol measurements were unconfounded and participants could be safely confronted with a psychosocial stress test.”

(6) P. 8, l. 146: Typo? Should this be 7.5% before the onset of their menarche?

Thank you for finding this mistake. It turns out, mothers’ and daughters’ percentages concerning their hormonal status were calculated wrongly. Everything should be in order now.

(7) P.9, ll. 168-170: Could the authors explain the passive observer role of the adolescents a bit further? What does it mean/how was it described to them that they should abstain from nonverbal communication? Obviously, this could also make them act unnaturally with regard to facial expressions. Suppressing emotional states, which may be necessary also, could also induce stress, other than the empathic stress aimed for in this research. I also wonder, whether parents were aware of the instructions given to their children. In the spirit of the TSST - probably not. Can the authors clarify?

Apart from this, "they were not allowed to.." may be better phrased as "they were instructed to not..." – at least I would hope this was true.

Thank you for raising this point. To clarify, parents were not aware of the detailed instructions given to their children, as adolescents were brought to the testing room before their parents. However, parents were instructed that their children would act as purely passive observers and would not partake in stress testing. Thus, most of the instructions given to the adolescents were also made transparent to their parents.

Regarding your query on abstaining from non-verbal communication and a possible manipulation of behavior and facial expressions, we absolutely agree. In detail, adolescents were instructed to “make no direct contact” with their parents. Emphasis was put onto them to act as “purely and silent observers”. They were *not* asked to refrain non-verbally communicating. Thus, we have changed the wording accordingly:

“Adolescents were asked to act strictly as passive and silent observers. As such, they were instructed to make no direct contact with either their parents or the TSST committee members.” In other words, we decided to give no specific instructions in how adolescents should act or what they were allowed to do. We expected them to understand this as not speaking to their parents, not directly gesturing at their parents etc., while giving leeway in terms of non-verbal communication. Still, instructions on how to act are very likely to alter behaviour, especially given that adolescents were also video-taped and the TSST situation itself was probably very unnatural to the adolescents. Overall, we believe that these instructions worked rather well, as no adolescent participant spoke to their parents during the TSST, while leaving the degree to which they non-verbally communicated up to them.

(8) P. 14, ll. 282-283; 282; “Because in our study adolescents were observing their parents, synchronization could only occur in one direction.”: Given that the adolescents were visible to their parents, this is not correct. The researchers, may, however only be interested in one direction of mimicry, and could justify that. However, it is very likely that mimicry also occurred in the other direction. In fact, the authors could easily check whether there was a difference between pacing and leading by also calculating parents lagging behind their children.

You are absolutely right, mimicry between adolescents and parents was likely bi-directional, since the two parties could look at each other, smile at each other, etc. We used the given phrasing because parents were the active part in the TSST, showing facial behaviour due to the stressor at hand, while adolescents passively observed. Given the active role of the parent and our research focus on empathic stress transmission, we chose to investigate only the adolescents’ mimicry. Accordingly, we have rephrased the statement in question as follows:

Page 16: “Because in our study adolescents were observing their parents, synchronization was only investigated in one direction. Although parents were also able to see their children and could have synchronized with their expressions, our particular interest was the degree to which adolescents mimicked their parents’ facial activity.” (9) P. 17, l. 339: I assume that the two factor solution was used based on the scree-plot. [Later] I have found this information in the results. Would be great if the authors stated this explicitly somewhere between ll. 338-339 to ease comprehension.

As suggested, we made this information more explicit earlier on in the methods section:

Page 19: “The number of factors in our EFA was calculated using parallel analysis (Horn, 1965) and diagnosed with a scree-plot. In parallel analysis, the eigenvalues of the factor solution are compared to the eigenvalues generated via Monte-Carlo simulation using random data (our parallel analysis scree-plot is shown in the Supplementary Results; Fig. S2).”

(10) For the regression models, were continuous predictors centered to their mean when including them in interaction terms? This would be indicated and should be corrected if this was not the case.

All continuous predictors were centered to assure interpretability of the model intercept and handle possible multicollinearity. This information can be found at:

Page 20: “Adolescent AUC_i and the mimicry factors were z-standardized to minimize multicollinearity.”

(11) P. 18, ll. 361-362: Can the authors give any insight into the reasons behind the high number of invalid HR recordings in adolescents? A good general rule to deal with missing values is to try to prevent them from occurring and there may be reasons why this failed in this instance?

In previous studies, we have had an acceptable number of invalid recordings with the breast belts used in this study. Although this is not a tested hypothesis, we believe that the variability in body size across the adolescents’ puberty stages lead to a far greater number of invalid HR recordings compared to their parents or earlier studies. The Zephyr-belts we used to measure a continuous ECG only came in two sizes, large or small. Our research assistants tried their best to adequately position the belts with enough pressure on the skin, assuring reliability between different body types.

Still, we were rather surprised on how much data was lost, given that adolescents were seated during the majority of the testing session. In future studies, we will utilize better alternatives to measure children’s and adolescents’ autonomic activity. For the current study, all ECG related results are interpreted as preliminary findings, given the unfortunate lack of power in our sample.

Results:

(12) P. 19, Figure 3: Please adjust the x-scale of the HR/HRV data to somewhat represent the timing of the study. This would help interpretation in context with the other measures. If this is not feasible, it could help if there were indicators for the start and end of the TSST in each subplot.

Thank you for this suggestion! We have changed the x-axis accordingly! We have also added an indicator for the start and end of the TSST.

(13) P. 20; WCLC: A common test for WCLC synchrony is, to check for statistical significance of the synchrony scores against some form of pseudo-synchrony. This check makes it able to distinguish interpersonal synchrony from synchrony that occurs by chance and in the case of this study also synchrony that occurs as a result of mere procedural aspects of the study. Examples for such pseudo-synchrony checks can be found in the extant literature (Ramseyer & Tschacher, 2011; Altmann, 2013; Riehle et al., 2017). A fairly easy solution could be to check for synchrony in video recordings of adult/adolescent combinations that are not parent and child, i.e. who did not actually interact with one another. Relatedly, Altmann (2013) used the problem that two participants respond to a similar stimulus and are in interaction with one another as rationale for his windowed cross-lagged regression. I am unsure if the interpersonal situation in the present study poses this exact problem, as the adolescents are not in the TSST situation themselves. Still, I would appreciate

if the authors dealt with aspects of synchrony that could occur for reasons other than empathic responding/mimicry somehow.

Thank you so much for raising this concern. The initial submission was absolutely lacking in controlling for pseudo-synchrony. We have now changed our approach from WCLC to WCLR after revisiting the relevant literature. As you pointed out, Altmann et al. (2013) as well as Schoenherr et al. (2019) provided ample evidence for a better fit of WCLR if working with cyclic data. We would suggest that activity of facial muscles is very much cyclic in nature, given that activation is always followed by deactivation. Thus, previous activity of a facial muscle will predict future activity, calling for the implementation of auto-regressive effects.

In a first attempt, the calculation of WCLR posed some concerns regarding calculation time, which we were able to circumvent by implementing linear regressions via C++ in R.

In the current revision, WCLR is now accomplished by comparing a simple auto-regressive model of the adolescents AU activity with a second model comprised of the auto-regressive effect and the parents' lagged AU activity. Only if the second model explains significantly more variance than the first model, the R^2 difference (R^2_{CC}) of the two models is retained and used in further calculation of peak picking and sync times.

According to Schoenherr et al. (2019) we now also log-transform our raw AU activity data and use a R_{CC}^2 threshold of >2.5 to further control for spurious correlations.

Please note that the resulting factor analysis yielded highly similar results as in our initial submission using WCLC.

(14) P. 20, WCLC: Apart from the correlations, it would be great if the authors could also present M (SD) for each AU - this is an important qualifier for synchrony interpretation. The results suggest, for example, that AU20 was synchronized for about 25% of the time, but this AU was probably not shown for 25% of the time. In fact, it is even good practice to control for both interactants' mean values when testing for effects of synchrony. This could be difficult in the present approach using the EFA, but one could try and extract latent factor scores for AUs' means based on the AU allocation to the two latent mimicry factors. At least presenting Ms and SDs in the supplement and interpreting this would be great for having that context.

It is absolutely correct, AU occurrence rates very much differed across AUs and participants. It also seems plausible to expect a relationship between the amount of AU activity and the time dyads spent synced, given that synchronicity can only occur if an AU is activated in both dyad members. Regarding your suggestion to control for AU means, we agree that our EFA approach does not easily allow to implement this. However, we also had some difficulty to understand your suggestion regarding an AU mean factor model. If you continue to perceive this issue as problematic, we would be very thankful for additional guidance regarding implementation. At present, we are grappling with the following questions:

Would you suggest to construct a factor model for AU means (average activity) using a similar factor structure as found for sync time EFA? If this should be the case, would you like to see whether a similar factor structure fits? Or whether there is a relationship between sync time factors and AU mean factors?

Currently, we have added two figures to the Supplementary Material showing the occurrence rates of AUs (Figure S5) and their mean intensities (Figure S6) for parents and adolescents.

(15) P. 21, ll. 405-406: In my opinion, the negative expression mimicry factor should be labelled only "synchronization with negative expressions". That it is comprised of various negative emotions does not necessarily mean that a synchronization with a complex expression has actually occurred.

Thank you for this suggestion. We very much agree and have changed the wording across the manuscript.

(16) P. 22, l. 416 and onwards: For all models that include a main effect of parental AUCi on adolescent AUCi, please also include the results for these main effects in the text and include these in the interpretation where adequate.

We have added all main results regarding effects of parental on adolescent AUCi into the main text. Please note that in light of changes made to the calculation of adolescent HRV in response to Reviewer 2 (i.e. using RMSSD instead of HF-HRV), our main results have changed. We no longer find a significant association between mimicry and empathic stress.

(17) P. 23, ll. 424-425; "The majority of adolescents peaked immediately after (40.3 %), 10 minutes after (23.4 %) or 20 minutes after the TSST (13.0 %).": Figure 3 suggests a steady decline in Cortisol in adolescents over time - on average... how does this fit with these results? Of course individual trajectories maybe cancelling out one another - but not if some 75% peaked in cortisol after the TSST. Would it not be fair to say that adolescents did not really show a Cortisol response?

Yes! It would absolutely be fair to say that adolescents did not really show a cortisol response and we have made this more clear in the results and discussion sections. The percentages given are based on the calculation of cortisol responder rates through baseline to peak change cortisol change scores.

Given that we have a high number of measurement points during the recovery phase (after the TSST), min-max reactivity tends to get "swallowed up" by stress recovery. Thus, we were attempting to convey another aspect of the stress response in our initial submission, that is, maximum cortisol secretion from baseline to peak. However, given average cortisol trajectories, we agree that this information was misleading. We have changed the wording throughout the results section accordingly.

Page 27: "Cortisol trajectories showed a decrease rather than an increase across the testing session, illustrated by 55.9 % of adolescents showing their highest individual cortisol measurement at baseline (see Fig. 3)." Page 32: "The relative lack of cortisol reactivity in the adolescents of this study, however, does not support the notion that their empathic stress [...]"

(18) P. 25, Fig. 6: The authors interpret this interaction as counter to their hypothesis, but is this not in line with the hypothesis, if mimicry was low and therefore did not facilitate

empathy and adolescents then did not respond with low HRV (high stress) themselves?

Since with the current changes to our data analysis and multiple testing approach the result in question is no longer present, we have removed all interpretations.

Discussion:

(19) P. 26, ll. 492-493; “one cannot frown (AU04) while at the same time raising one’s eyebrows (AU02).”: I am sure that some people can and would advise against such absolute statements in favor of more probabilistic ones.

Thank you for raising this concern. With this statement, we were hoping to illustrate the natural correlation between certain muscles in the face, which has to be accounted for in facial expression and mimicry analysis. As you suggested, we have altered the absolute stance of our argument.

Page 29: “At the same time, specific AUs act in opposite directions. For example, it is highly unlikely to frown (AU04) while simultaneously raising one’s eyebrows (AU02).”

(20) P. 26, l. 498; “corrugator fascilii”: This one is more commonly known as Corrugator supercilii.

Thanks for pointing this out, we have changed it throughout the manuscript.

(21) P. 27, ll. 508-510, sentence on facial feedback hypothesis: I disagree with this statement, because it insinuates that the findings were evidence of a causal relationship between mimicry and affective-physiological state. The data do not prove the temporal relationship of this putative causal direction (i.e., it is not clear if mimicry/emotional expression caused the internal affective state or if the affective state caused emotional expression) and the data also do not show that there is a causal link, even if we assumed this temporal correlation to be in line with the putative causal direction. It could be, for instance, that shared knowledge about emotions and mentalization (i.e. cognitive portions of empathy) cause internal affective states in both the parent and the adolescent, which both of them then express similarly on their faces based on shared display rules. Now - I do consider some of the findings in line with the hypothesis of a facial feedback or mimicry being a potential causal factor driving this. However, before one may call this causal, more (experimental) research needs to follow. As this interpretation is appearing in several ways in the discussion, I would advise to revise the discussion and tone down these claims of causal directionality and causality. The concluding sentence in the abstract should also be revised accordingly.

We apologize for the wording used in the discussion and completely agree that our data is not suitable to infer any causal claims between empathic stress sharing and mimicry.

Accordingly, we have removed any wordings suggesting a causal relationship. Given that we no longer find a significant association between empathic stress and facial mimicry, no further discussion of facial feedback as a possible explanation is presented.

(22) P. 28, ll. 537-539; “Actual interaction between adolescents and parents rather than passive observation may thus offer the possibility for bi-directional emotional mimicry, thus boosting empathic stress occurrence.”: Again, I would argue that back-communication occurred on some scale - otherwise the authors would not have been able to detect facial expressions in the adolescents.

Writing this comment brings me to another point: Is there any way to quantify the reliability/internal consistency of the OpenFace AU detection?

As we have stated above, we very much agree with your assumption that bi-directional emotional mimicry was possible and have changed this throughout the manuscript (methods and discussion). However, we would uphold our statement that the observation of the TSST was not a direct interaction between parent and adolescent. Accordingly, we have rephrased the sentence as follows.

Page 16: “Because in our study adolescents were observing their parents, synchronization was only investigated in one direction. Although parents were also able to see their children and could have synchronized with their expressions, our particular interest was the degree to which adolescents mimicked their parents’ facial activity.” Regarding your question on the reliability/internal consistency of the OpenFace AU detection, we would like to refer you to the extant literature and the very detailed documentation of OpenFace (<https://github.com/TadasBaltrusaitis/OpenFace>). As with any deep neural network, we would expect that reliability mostly depends on the amount and variability of faces/situations the model was trained on, which to our knowledge has shown good reliability for Western European data sets. Quantifying the internal consistency of the AU activation measures sounds like a very good idea, but at this point we are not sure (1) how we would do this and (2) whether this would fit within the scope of the current manuscript. However, if you have any ideas on how to incorporate a reliability/internal consistency check in the current context, we would be happy to oblige!

We have also commented on the reliability and validity of our overall approach in the discussion:

Page 32: “Last, the accuracy of our approach to measuring facial mimicry using video-based AU detection (openFace; Baltrušaitis et al., 2016), and WCLR remains uncertain. Since this method combination is novel, its validity and reliability still need to be tested. Consequently, it remains unclear to what extent our latent factors accurately reflect facial mimicry between parents and adolescents.”

(23) P. 29, l. 547-548; “Recruiting fathers next to mothers increased generalizability of findings to households with heterosexual parents.”: I do not understand what this has to do with the question of whether or not the household includes hetero- or homosexual parents? The sampling increases the generalizability from only mothers to also fathers, which is a noteworthy strength, but this is irrespective of sexuality.

Absolutely correct. We have changed the statement accordingly:

Page 33: “Recruiting fathers next to mothers increased the generalizability of our findings.”

Conclusion:

(24) P. 29, ll. 560-561: Here, conclusions are drawn based on the whole set of hypotheses but based on some singular positive in light of several negative findings. This would constitute a case where FWE-correction is indicated even according to García-Pérez (2023).

You are absolutely right

We have changed our multiple testing approach and are now correcting for clusters of stress markers. In consequence, the significance threshold has been increased to $p = 0.0167$.

Page 20: “**Multiple testing.** Because our hypotheses regarding the effects of mimicry and socio-emotional capacities on empathic stress were tested across overall six stress markers, a Bonferroni correction was applied. The significance threshold was adjusted to $p = 0.0167$ ($0.05/3$) reflecting two groups of three physiological (cortisol, HR, HRV) and three psychological (STAI, EC, PD) stress markers. Previous studies have shown little to no covariance between physiological and subjective stress reactivity, illustrating the complexity of the human stress response (Engert et al., 2018; Engert et al., 2011). Because physiological or psychological empathic stress is thus interpreted as a specific rather than general case of empathic stress, no further correction was applied (García-Pérez, 2023).”

Consequently, we do not find significant associations between facial mimicry and empathic stress:

Page 33: “In summary, our findings showed that emotional mimicry during the TSST was not associated with stress transmission from parents to their adolescent children. These null-findings could possibly be explained by the overall lack of a significant empathic cortisol stress response in adolescents, the decision to measure mimicry only during the TSST rather than when parents and adolescents are reunited following the TSST, and the novel approach used to calculate emotional mimicry employing video-based AU detection.”

(25) P. 30, l. 568; “shared on a sub-conscious perceptual level”: I do not think that this study has necessarily tested mimicry as a sub-conscious construct and, as mentioned above, one could even argue the necessity of a perceptual aspect of what has been assessed (i.e., anticipation processes based on generalized expectations play into the expression). Maybe this could be stated more neutrally.

Thank you for this suggestion. Given that we no longer find any association between mimicry and empathic stress, we have removed this interpretation from the conclusion completely. Instead, our conclusion not focusses more on our novel methodological approach.

Page 33: “In summary, our findings showed that emotional mimicry during the TSST was not associated with stress transmission from parents to their adolescent children. These null-findings could possibly be explained by the overall lack of a significant empathic cortisol stress response in adolescents, the decision to measure mimicry only during the TSST rather than when parents and adolescents are reunited following the TSST, and the novel approach used to calculate emotional mimicry employing video-based AU detection. We conclude that while the usage of digital biomarkers such as AU activity appears to be a promising tool in the study of emotional mimicry, further studies using a similar approach are essential to establish the validity and reliability of the methods employed. Especially the feasibility and cost-efficiency of video recordings compared to bio-physiological measures such as

electromyography provide a compelling argument to expand this line of research. Given that “one cannot not communicate” (Watzlawick et al., 2017), non-verbal communication must be recognized as a crucial research avenue, not only in the transmission of psychosocial stress but in human interaction as a whole.”

Reviewer #2 (Remarks to the Author):

The manuscript outlines a study that was designed to measure facial mimicry between parent and adolescent while the parent was being stressed. The goal of this study was to elucidate a mechanism by which stress can be transmitted from one individual to the next. The authors of this study recruited 76 dyads to the laboratory. Cortisol, heart rate, HF-HRV and a number of subjective stress measures were used in addition to being videotaped to assess facial mimicry.

First, I would like to point out how impressed I am with the methods of this study. Synchrony work can be extremely difficult and often end up in the weeds, so to speak, but as I read through this manuscript I was thoroughly impressed and satisfied with all methodological choices made by the authors. So on this note, job well done to the authors who made well-informed and discerning choices throughout the analyses.

My main critique of this work is twofold:

1) HF-HRV plays a centrepiece role in this paper – yet its calculation was likely done improperly. HF-HRV in adolescents must be calculated using a different frequency spectrum due to the heightened breathing rate of adolescents, as compared to adults. For example, a typical adult breathes 9 breaths/min while a 12-year old breathes at 14 breaths/minute and a 16-year old at 11 breaths per minute. Within the methods section, the authors did not specify the frequency window used when assessing HF-HRV (usually is between 0.15 and 0.40 Hz for adults). Firstly, this needs to be specified. Secondly, due to the heightened breathing rate of adolescents, this frequency window needs to be modified for the adolescent participants based on their age. So for example HF-HRV in a 12 year old wouldn't be $>0.15\text{Hz}$, it would be $>0.23\text{Hz}$ instead. These analyses need to be redone accordingly.

Alternatively, the authors can shift their analysis to using RMSSD. This is not a frequency-based measure of HRV, and thus is independent of breathing rate. RMSSD would not need to be corrected for due to age, therefore yielding it as a potentially superior measure of HRV in this specific context, where HRV is needed to be compared across generations.

I am also a bit perturbed by the massive drop in n due to issues with the cardiac data.

Dropping 34 dyads (out of 76) is dropping 45% of the data. I would like to see more than one line dedicated to this issue. Understood sometimes data is messy and cannot be used, or there are technical difficulties, but 45% is outside of the norm, and at least an extra line or two explaining this would be helpful.

Thank you so much for raising this concern! As you suggested, we have changed our HRV analysis to using RMSSD.

Furthermore, we now elaborate more on the ECG data dropout. Unfortunately, next to the dyads that were dropped due to missing ECG data, there were several dyads that had missing video (but intact ECG) data, further limiting the ECG data set.

2) My second main qualm is the title and conclusion of this paper which states that mimicry boosts empathic stress responses. While this paper has very interesting findings (I was enthralled!) – I did not see this as a takeaway from the paper. It certainly would have made for a neat picture, but facial mimicry did not seem to be related to any of the subjective stress measures except for the STAI, and this was in line with the vicarious stress model, not the

stress resonance model. So my takeaway from this is that adolescents who experienced more anxiety watching their parents get stressed, had more facial synchrony with them (for negative expressions). The directionality of this isn't necessarily obvious – couldn't it be that adolescents who are more emotionally reactive (scoring high on STAI while watching a stressed parent) experience more facial mimicry? Facial mimicry was related to HF-HRV (results tbd – maybe they will become more intuitive with age correction) but as of now, the results yield an intriguing relation, but not one that supports the claim that empathic stress has been boosted. If anything, I believe the interesting takeaway from this paper is that facial mimicry isn't more related to each of the measures (cortisol synchrony, heart rate synchrony etc.) That being said, the authors chose to use AUC measures for heart rate and HF-HRV – synchrony of these measures could be done in a more fine-scaled way. I can understand why that would overload this paper – its analyses are already very hefty, but it would certainly be interesting! Basically, my feedback is to tone down the conclusions of the paper and keep them more in tune with the actual results. I don't think that takes away from the clearly impressive package that this paper is.

Thank you for raising this very valid point, and encouraging a more cautious interpretation. We agree that we simply cannot make causal inferences regarding the link of mimicry and empathic stress, and thus ended up with the wrong conclusions. Since with the various changes introduced (1) replacing the WCLC by a WCLR approach, 2) using RMSSD instead of HF-HRV, and 3) adding multiple testing correction), the link between mimicry and stress transmission is no longer present, no further changes in interpretation needed to be implemented in the discussion section.

To quickly comment on your suggestion for more fine-grained analyses of ECG data. Yes, we have been thinking about looking into e.g. facial synchrony in relation to heart-rate synchrony, which both could be investigated using very fine grained correlational measures (such as WCLC or WCLR). But as you already commented, in the current paper, we wish to focus on the measurement of facial mimicry in the context of an empathic stress task. Therefore, we tailored this manuscript towards a more methodological piece on facial mimicry in empathic stress transmission.

Minor issues:

1) In the abstract it says $n = 77$, and in the manuscript it says $n = 76$, which is it?

This was a typo and has been changed. Overall, the study included 77 dyads.

2) Definition of empathy presented in first line is more so a definition of affective empathy than an all-encompassing definition of the phenomenon.

Thank you for pointing this out. We have changed the introduction to more clearly differentiate between cognitive and affective empathy.

Page 3: “Beyond the sharing of subjective experience, empathy entails the reproduction of others' physiological and neural states. This can occur through affective empathy, which is the mere reproduction of another's feeling, or cognitive empathy, which combines affect sharing with mentalizing about another's thoughts and feelings (Batson, 2009).”

3) Line 65 ends without a citation, and it is unclear if they are referencing Miller et al. 2013 or Engert et al., 2014

Thank you for pointing this out - we have added the respective reference to more clearly distinguish between the sources mentioned.

4) The concept of “action units” was not talked about enough in the discussion. In line 103 when it is mentioned it is unclear what the authors are talking about. I would find a space earlier in the intro to write a line or two about action units and what they are.

Thank you also for this suggestion. We have added more information on action units in our introduction section.

Page 6: “The FACS constitutes a taxonomy to categorize movements of facial muscles or muscle groups in terms of action units (AUs). For example, AU12 describes the action of lip corner pulling, a facial movement governed by the zygomaticus major muscle.”

5) I would refine the last line of the introduction (“Based on the role of mimicry..”) it’s unclear and can be better written

Absolutely. We have rephrased the last line of the introduction as follows:

Page 6: “In the current study, we investigated subjective-psychological and physiological empathic stress and its association with emotional mimicry in adolescents observing either their mothers or fathers undergo the TSST. Considering the multi-modality and complexity of the human stress response (Engert et al., 2018), adolescents and parents simultaneously provided measurements of cortisol release, sympathetic, parasympathetic, and subjective stress reactivity. Additionally, dyads were videotaped during the stress task to gauge adolescents’ emotional mimicry via shared AU activity. Based on the notion that mimicry acts as a sub-conscious precursor of affective and cognitive empathic processes (Batson, 2009), we expected greater adolescent stress-resonance and vicarious stress given a higher degree of emotional mimicry between parents and adolescents”

6) Dyads rested for 40 minutes. What did they do in that time? Were they allowed to talk? Did they have reading materials?

Dyads were allowed to freely interact with each other before and after the TSST. They were allowed to bring their own reading material, except work- or school-related things. They were not allowed to use their smartphones. A very small variety of reading material was also provided by us. We have added this information to the manuscript to more clearly reflect how dyads spent their time before and after the TSST.

Page 10: “They were encouraged to bring their own reading material as pastime, except for material related to work or school. Participants were asked to switch off their smartphones during the entirety of the testing session.”

7) Line 194, when the TSST is introduced, be sure to indicate that it’s been shown to elicit a robust stress response

Thank you for pointing this out, we have now added this information.

Page 11: “The TSST has been shown to reliably elicit a stress response in about 80 % of adult participants (Kirschbaum et al., 1993; Kudielka et al., 2007).”

8) Line 235; I wouldn't mind seeing one or two examples of the questions or adjectives used in the ERS, for those unfamiliar with it

We have added examples for both the personal distress and the empathic concern scale of the ERS.

Page 13: “The ERS prompts feelings for another individual on six adjectives measuring empathic concern (e.g. “softhearted“, “compassionate”) and eight adjectives measuring personal distress (e.g. “troubled”, “worried”) on a seven-point Likert scale from 1 (not at all) to 7 (very much).”

9) Why isn't AU 25 included in Fig. 2?

AU25 describes the act of opening your mouth. The illustrator decided to refrain from including it because it is (1) rather self explanatory and (2) would have resulted in 16 AUs, breaking the figures symmetry with 3 rows of 4 AU. (3) It would look very similar to AU26 (Jaw Drop). If you would like to have AU25 included in Fig. 2 we would be happy to change this figure in second a revision.

10) I like how level of synchronization was operationalized by time in sync...but what about the actual level of the correlation? Why was that not looked at at all?

This is a great question. First, please note that due to comments on pseudo-synchrony presented by reviewer #1, we have changed our approach to windowed-cross-lagged-regression (WCLR). Hence, we are now looking at regression slopes regarding the strength and direction of the association. Please see the following paragraph in our methods for an in-depth introduction to WCLR:

Page 16: “Given a certain window at a specific lag (e.g. window = 1-125, lag = 2), WCLR then compares whether the amount of variance explained by the partner (i.e., the parent) exceeds that of an auto-regression (i.e., the adolescent's previous activity). Thus, auto-correlation existent in the lagged signal is controlled for. In detail, two models are fitted and compared at each step (Altmann, 2011, p.3).

$$\text{Model 1: } X_{1t+\tau} = \beta_0 + \beta_1 X_{1t} + \varepsilon_{1t}$$

$$\text{Model 2: } X_{1t+\tau} = \beta_0 + \beta_1 X_{1t} + \beta_2 X_{2t} + \varepsilon_{1t}$$

In our case, X_{1t} pertains to the data points of the adolescent and X_{2t} pertains to the data points of the parent at a time t . Model 1 is a simple auto-regressive model, in which the adolescent's data points τ steps further ($X_{1t+\tau}$) are explained by their initial data points ($\beta_1 X_{1t}$).

Model 2 includes the auto-regressive effect ($\beta_1 X_{1t}$) and the effect of the parents ($\beta_2 X_{2t}$).

To gauge an index of cross-correlation irrespective of autocorrelation, R^2_{cc} is calculated as:

$$R^2_{cc} = R^2_{\text{Model 2}} - R^2_{\text{Model 1}}$$

To further assure that the cross-correlation is not random, the R^2_{cc} is only retained if model 2 explains significantly more variance than model 1. For a more detailed description of WCLR, please see the relevant literature (Altmann, 2011; Altmann, 2013; Schoenherr et al., 2019).”

Operationalizing facial mimicry as a measure of time in sync offered a fitting comparative index that also includes the strength of association between dyad members, due to the fact that a higher mean correlation between signals will lead to a higher amount of time in sync.

To offer a more fine-grained illustration of our attempt to measure facial mimicry via WCLR, we have now added descriptive information on the WCLR results. This includes the mean lag at which sync intervals converged and mean R_{cc}^2 in the extracted sync intervals across AU (see Table 1 in the Results and figures S7 and S8 in the Supplementary Results.)

To add on, we have also tried to more clearly convey that our measure of sync time is not a direct measure of association strength. But given that correlations in the sync intervals had to be high because of the peak picking procedure including an R_{cc}^2 threshold, an indirect measure of association strength is included.

Page 19: “While sync intervals will include comparatively high correlations due to the nature of peak-picking, a sync interval does not necessarily entail an intense facial expression in both target and observer.”

11) Why were sync intervals shorter than 1 second excluded?

This pertains to the methodology of sync interval calculation introduced by Altmann (2013). We strictly adhered to Altmann’s suggestions, which was to drop sync intervals shorter than 1 second (25 frames in our data). Given that in this study facial mimicry rather than body movement synchrony (as in Altmann’s examples) was investigated, we agree that excluding sync intervals of 1 second or less might be too conservative. Accordingly, we have changed our approach to to include sync intervals of at least 10 frames (0.4 seconds). This criterion has previously been used in another publication by the Altmann group employing sync interval estimation (Schoenherr et al., 2018).

Page 19: “Sync intervals shorter than 10 peaks (0.4 second) were excluded as suggested by Schoenherr et al. (2018).”

12) Figure 3 has inconsistent x axes – can they all be in minutes? That would be more intuitive

Absolutely. We have changed the x axes of the autonomic markers to represent minutes rather than the time phases. We have also added an indicator for when the TSST actually took place.

13) I can safely assume that the 34 missing dyads only affected the HF-HRV and HR data and all dyads were included in all other analyses, right?

Yes. We have added a table in the Supplementary Results (Table S1) that shows the amount of dyads included in each analysis and the percent of data imputed.

14) Typo Line 491 (lip corner not lip cornor)

Thank you – fixed,

15) Line 561 “synchronizing with their caregiver’s facial activity increased their empathic stress reactivity” – did it? From my reading I didn’t see this, and this harkens back to the second main issue I brought up above

With the changes made to our analysis approach, these findings are no longer present. The mentioned faulty interpretation has been removed.

Kudos:

- Using audio waveform to synchronize the videos is very smart! Nice work
- window sizes, lags and increments were all nicely chosen
- Bonforonni at $p < 0.00000006$ is great. From my calculations may even be an overcorrection? But nice work being conservative
- Loved the idea of using an EFA to derive to latent factors from the Aus
- Love the inclusion of Fig. 3 to see the dynamics of all these measures across dyads – many papers seem to omit such basic graphs but they really add to the manuscript
- Interesting to see the finding that on average 24% of the time was spent with facial mimicry! Just this finding alone is laudable given the extensive analyses needed to arrive at this number
- well-written and clear
- great choice of statistical methods and excellent word clearly spelling them out

Overall I was especially impressed with this paper but my two main issues are that the HF-HRV data must be age-corrected or changed to RMSSD and the discussion/title abstract should be edited to better represent the actual findings of the paper (which may change with the HF-HRV analyses). I think this paper can be influential to the field in showing a very well-laid out method of analysing synchronous facial muscle activity from video footage; this opens the door to many interesting avenues in the empathy domain.

Again, thank you so much for this critical and supportive review of our manuscript! Regarding the HF-HRV measurements in the adolescents and our result interpretations, we hope to have sufficiently adressed all your concerns.

Reviewer #3 (Remarks to the Author):

The current paper is concerned with the spontaneous effect of stress spillover, specifically, the role of mimicry in the transmission of stress. The authors examined a sample of adolescents who observed their parents undergo a stressful situation. Looking at young adults is particularly interesting as it fills an important gap in the literature. Similarly, looking at mimicry is an interesting way to quantify contagion, and in particular contagion in familiar dyads. Overall, this study is a timely and important contribution to the literature that will help us to better understand the link between stress and close relationships.

I first want to congratulate the authors for an interesting and engaging manuscript. The methodological aspect of the studies is expertly done and in line with the high standards for stress research. As such, the manuscript has many strengths and could make an important contribution to the field. The main drawback is the small sample size for the autonomic markers—the ones that are statistically significant. This issue must be addressed more clearly and even added explicitly to the abstract. In addition, I would like a more thorough discussion of the relatively small stress response in the observers. This seems to deviate from previous studies (also from the same group using a similar design). This is not a problem per se, but it is curious/interesting enough that it should warrant some attention. In my opinion, the results (and rigorous method) still warrant a possible publication, despite this. There are some additional issues that I want to highlight to improve the paper before I can recommend it for publication. Together with some more general comments I have added them below.

Comments:

There is a recent review (Nitschke & Bartz, 2023; Neuroscience and Biobehavioral Reviews) that has reviewed the evidence of empathic stress (or stress contagion) that goes a bit beyond the (admittedly excellent) summary by Engert et al. (2019) by discussing the importance of familiarity (or closeness) for the spillover effect in more detail.

Thank you for pointing this out; we have added the review to our introduction.

I greatly appreciate the transparent preregistration and the open data. This is excellent and should be acknowledged as such.

Thank you so much!

Sample size. It is unclear how many participants were excluded due to various issues mentioned in the manuscript. There should be a table in the Supplemental Materials indicating the exact participant/dyad number for each DV.

We have added a table showing the existing sample for each DV to the Supplementary Material (Table S1). This information can also be found in the respective regression table (table S4). Last, we have added the amount of dyads included in each regression model in the section on missing data in the methods section:

There also seems to be a mismatch between parents ($n=76$) and children ($n=77$). In addition, the percentage reporting for mothers is confusing and would benefit from additional information (i.e., 15.8 % of all parents; 9.5% of mothers). I also don't know what phase of the

menstrual cycle the 28.5% of mothers were in that was not reported (same with daughters—the numbers do not add up).

We apologize for the confusion. The complete sample included N=77 dyads. Quite a few mistakes were spotted in the section on descriptive information (as also pointed out by reviewer #1), all of which have been corrected in the revised manuscript. We have also added the relevant group (e.g. X % of mothers) where necessary.

I am not sure that providing all participants with orange juice and chocolate will equalize their blood sugar levels—specifically, also because this “equalization” was not tested. In my opinion, this approach introduces more confounds than a spontaneous, unaltered HPA axis response. Yes, a manipulated HPA axis response might look nicer on paper, but there is insufficient evidence to attribute this spike in cortisol solely to a higher metabolic potential. Of this is to say, this approach makes studies less comparable and glucocorticoid-specific effects less confident. I do not know if this approach standardizes metabolic output potential. That said, I appreciate the disclosure, and this transparency is sufficient to address methodological differences in stress induction, particularly in the glucocorticoid stress response between studies. I do not need this to be addressed any further.

Thank you for pointing this out. We will be mindful of this critique in future stress studies conducted in our lab.

All other aspects of the stress protocol of the study are expertly done.

Thank you.

In regards to the handling of the cortisol data. The skewness of raw data is not very important for linear models (or not important at all) as it is rather about the skewness of the residuals from the model. As such, it is not necessary to preemptively transform the raw data by applying a log transformation. Ideally, the authors should check if including raw cortisol data in the models violates the homoscedasticity assumption. Similarly, it might be a good idea to run the models without winsorisation first. In my opinion, stress manipulations will naturally produce extreme values—and most of these extreme values will carry important information that should not be preemptively dismissed (or reduced). I do not need the authors to change any of their analyses, but, at the very least, the authors should provide numbers on how many data points were altered.

You are absolutely right that normality of the residuals of a model rather than a normally distributed DV has to be given to adequately interpret the results of our multiple linear regressions. We chose to apply log transformations as well as winsorization because this approach has been used regularly in previous submissions.

Based on your comment, we have changed this approach such that only log-transformed values were used if model residuals were skewed. This was the case only for the HRV measure. All other linear regression models are now calculated using the raw stress marker values. No winsorization was applied.

Page 13: **“Stress marker data preparation.** RMSSD HRV data were transformed via the natural logarithm because subsequent model residuals were non-normally distributed. To

attain a measure of parental and adolescent stress sensitivity, the area under the curve with respect to increase (AUC_i; Pruessner et al., 2003) was calculated for both dyad members and each stress marker (cortisol, heart rate, HRV, subjective stress). Because state empathic concern and state personal distress were only measured in adolescents, AUC_is were only calculated for them.”

Apart from the previous comment, I do not have any issues with the statistical approaches. They appear to be expertly executed.

Thank you, we really appreciate this positive evaluation.

I understand the rationale for not correcting for multiple comparisons. Still, I find it a bit odd as a blanket statement since some dependent variables clearly measure similar things—e.g., in the case of, autonomic activation, there are two measures: HR and HRV). Otherwise, I tend to agree.

We have changed our multiple testing approach and are now correcting for clusters of stress markers. In consequence, the significance threshold has been increased to $p = 0.0167$.

Page 20: “**Multiple testing.** Because our hypotheses regarding the effects of mimicry and socio-emotional capacities on empathic stress were tested across overall six stress markers, a Bonferroni correction was applied. The significance threshold was adjusted to $p = 0.0167$ ($0.05/3$) reflecting two groups of three physiological (cortisol, HR, HRV) and three psychological (STAI, EC, PD) stress markers. Previous studies have shown little to no covariance between physiological and subjective stress reactivity, illustrating the complexity of the human stress response (Engert et al., 2018; Engert et al., 2011). Because physiological or psychological empathic stress is thus interpreted as a specific rather than general case of empathic stress, no further correction was applied (García-Pérez, 2023).”

The authors should include how many data points (percentage) were imputed. It is also important to include the exact method (including packages if conducted in R).

We used the mice package for multiple imputation and have added the relevant reference. We have also added the amount of data points imputed in percent in the Supplementary Results table S1.

“Percentages of values imputed for each stress marker can be found in Table S1 of the Supplementary Results.”

The power analysis needs further clarification. The (updated) pre-registered power analysis seems very reasonable and estimates that “a minimum sample size of $N = 64$ dyads is required” to find a moderate effect ($f^2 = 0.1$). This is more or less in line with the power estimation here. However, noticeably, the sample reported on ($n=76$; not sure how many/ or any were excluded) is smaller. I understand that the authors accounted for drop-outs in their OSF pre-registration, but this is not very well reported in the manuscript. The authors should clarify this and also make a short assessment of the current sample regarding the power

analysis.

Thank you for raising this concern. This was an error on our side. The power-analysis in our manuscript was calculated for a two-sided t-test, while the power-analysis in our preregistration was calculated for a one-sided t-test. Given that we have directed hypothesis (e.g. cortisol secretion in adolescents is expected to be greater rather than lower than zero) a one-sided t-test of regression coefficients seems plausible. Thus, we have corrected the power-analysis in our manuscript:

Furthermore, we have added a post-hoc power analysis for each of our DVs (depending on the drop-outs present) given an effect size of ($f^2 = 0.1$).

It would be helpful to include the time of observation in the figures (including the durations), as is, this is not very informative, especially given the different x-axis, and would benefit from more information.

We have changed figures 3C and 3D according to this suggestion. They now include the exact time (and thus the duration) of each phase. We have also added an indicator for when the TSST took place in the context of the overall testing session.

For the Supplemental Materials, the table would benefit from additional information (i.e., notes at the bottom) to explain the variables.

We have added notes to all figures and tables to explain variables in the Supplementary Results (Table S1).

Overall I like the points from the discussion and the interpretation of the findings in the context of other research. However, the discussion is a bit disjointed and would benefit from a thorough readthrough. All the ideas are there, but they do not necessarily build on each other. I also wish that some of the ideas (or cited research) were described in more detail, as it would help to put the research in context. I am also missing a big-picture assessment of the findings. For a more general journal, such as *CommPsych*, this is necessary as not every reader will be as in the weeds of stress research as the authors (or the reviewers), it would help make the findings more accessible (this could also benefit the intro, to some extent).

Thank you for pointing this out, we totally agree. In light of our re-analysis including a windowed-cross-lagged-regression (WCLR) approach, using RMSSD as a measure of HRV and the application of multiple testing correction, the results pertaining to empathic stress and mimicry have changed dramatically, that is, they are no longer significant. As such, we have largely re-worked our discussion and have put greater emphasis on our novel approach to measure video-based mimicry during an empathic stress test. We now also provide more general implications of our study, as required from a journal such as *CommPsych*.

Page 33: “Especially the feasibility and cost-efficiency of video recordings compared to biophysiological measures such as electromyography provide a compelling argument to expand this line of research. Given that “one cannot not communicate” (Watzlawick et al., 2017), non-verbal communication must be recognized as a crucial research avenue, not only in the transmission of psychosocial stress but in human interaction as a whole.”

I would focus more on the fact that empathic stress was not very high in this adolescent sample. This may be due to the sample, the nature of the age of participants, and some methodological choices. It might also be important to put this finding in context, notably, how many adolescents showed a stress response (or empathic response)? Can this be quantified similarly to previous papers on this topic (e.g., Engert et al.)? For example, for Cortisol, studies on stress contagion have shown increases in 7%-40% of the observer sample (see Nitschke & Bartz, 2023). For ANS activation, neither Buchanan nor Engert has found meaningful co-activation, but it appears that there (but small and de-synchronized) increase in HR in the current sample of adolescents. The fact that subjective stress levels are strongly co-activated in both observers and targets should also not be dismissed. It further highlights the complexity of the stress response.

These are some great suggestions. We have added the proportion of significant cortisol responders in our adolescent sample to the results and discuss the overall lack of empathic stress in the parent-adolescent dyad in far more detail.

Page 22: “N = 9 (11.7 %) adolescents showed significant cortisol release of > 1.5 nmol/l. In detail, N = 5 daughters and N = 4 sons were categorized as cortisol responders. Regarding the type of parent that was watched, N = 7 or 17.1 % of adolescents watching their mother and N = 2 or 5.55 % of adolescents watching their fathers were categorized as cortisol responders.”

Page 30: “Overall, empathic stress transmission from parents to their adolescents was lacking in the current sample. This was evidenced by declining cortisol trajectories over the testing session in adolescents, although parents showed a relatively high proportion of cortisol responders, suggesting successful first-hand stress induction. In the adolescents, cortisol responder rates were comparatively low, with only 11.7 % exceeding the 1.5 nmol/l threshold. Previous studies have found empathic responder rates of up to 40 % in romantic couples (Engert et al., 2014) and 17.9 % in 8 to 12-year-old children observing their mothers in the TSST (Blasberg et al., 2023). Consequently, the empathic responder rate observed in the current adolescent sample is closer to that of strangers (10 %; Engert et al., 2014). Given the overall decline in cortisol levels, but a time-lagged reaction in autonomic activity and a pronounced subjective stress response in the adolescents, we suggest that rather than empathic *stress*, empathic *arousal* was captured.

We suggest that the lack of empathic cortisol responses in adolescents was driven mainly by the relationship under investigation. First, cortisol responder rates of adolescents observing their fathers (5.55 %) were lower than of those observing their mothers (17.1 %), which were comparable to those of middle-aged children observing their mothers (17.9 %; Blasberg et al., 2023). Maternal stress may be more familiar and relevant to adolescents than paternal stress, because although allocation of primary care-giving has shifted towards a more balanced distribution of fathering and mothering (Radin, 1994), socially constructed roles remain prevalent. Fathers are often viewed as protectors, while mothers continue to be seen as the nurturing parent (Pakaluk & Price, 2020). This is evidenced by persistent behavioral differences, with fathers spending more working hours and mothers investing more time in child-care (Bianchi & Milkie, 2010). Importantly, this dependence on primary care giving may not be sex- but rather role-specific.

A second possible explanation may be the extent of helping behavior that is expected of adolescents. In general, triggering helping behavior may be one of the main functions of empathic stress occurrence (Engert et al., 2019). However, since children are typically not responsible for alleviating their parents' stress, stress-transmission might be perturbed rather than boosted in this specific relationship. Compared to younger children, it may be even less likely to occur in adolescents striving for independence from their parents (Allen, 2008)."

It has also been reported that the corrugator (and maybe frowning more generally) might be less reactive in social contexts (Hess & Fischer, 2013; Nitschke et al., 2019; Niedenthal et al., 2010; Carr, 2014), and it might therefore be difficult to pick up on more subtle changes.

Thank you for raising this point, which we have added to our discussion:

Page 30: "However, the corrugator has shown weak associations in terms of emotional mimicry (Lamm et al., 2008; Riehle et al., 2017) and might be less reactive than other muscles during social interaction in general (Hess & Fischer, 2013; Niedenthal et al., 2010; Nitschke et al., 2020). Hence, the current study produces further evidence that specifically frowning is not as readily reproduced as other facial expressions."

In that regard, the reported effect by Nitschke et al., (2020) was specific to zygomaticus activation—stress reduced reciprocal smiling, while it did not affect corrugator activation. Importantly, here the observers were actively stressed, whereas the target was unstressed. As such, the data might be especially informative for adolescents who were particularly stressed by observing their parents (i.e., AUCi-adolescent—mimicry-smiles).

Thank you for clarifying this. We have added the respective study details to the discussion:

Page 31: "None of the investigated stress markers were found to be associated with emotional mimicry. Mimicry specific to smiling could have been perturbed by increases in empathic stress, given recent evidence of a negative association between acute psychosocial stress and zygomaticus mimicry (Nitschke et al., 2020). In their study, participants zygomaticus and corrugator mimicry were measured in reaction to videos using EMG at a baseline (outside of stress) and after psychosocial stress induction via the TSST. They found that increasing levels of cortisol were negatively associated with reciprocal smiling, suggesting reductions in zygomaticus mimicry during acute psychosocial stress. The relative lack of cortisol reactivity in the adolescents of this study, however, does not support the notion that their empathic stress contributed to reductions in facial mimicry or vice versa. Rather, adolescents showed little to no significant empathic stress reactions but rather decreases in HRV as well as increases in heart rate and subjective stress experience."

Dear editor and reviewers,

we would like to thank you for all of your constructive comments and suggestions. Following, we will address the further changes asked for by reviewer #1:

Reviewer #1 (Remarks to the Author):

I want to congratulate the authors on such a thoughtful and constructive revision of their manuscript, which has now significantly gained in strength. I do have three comments left pertaining to the WCLR analysis. These may not warrant an entire revision cycle, however:

Thank you so much!

p. 18: Typo in “We chose similar parameters as Riehle et al. (2017), the currently one other study investigating facial mimicry using WCLR.”: We actually used WCLC, not ...R in this paper.

Thank you for pointing this out! We have corrected this typo!

p. 18: In “Subsequently, all negative slopes and slopes with an $RCC^2 < 1$ were dropped from the RCC^2 matrices.”: Is this value (< 1) correctly stated? R^2 would not increase beyond 1, or am I missing something?

No! This is a typo. We have fixed this accordingly:

Page 18: “Subsequently, all negative slopes and slopes with an $R_{CC}^2 < 0.25$ were dropped from the R_{CC}^2 matrices.”

WCLR analysis: The authors now have used an absolute R^2 threshold of 0.25 for their synchrony measure based on Schoenherr et al. (2019). It could be confusing that the authors also state that the significance of synchrony is determined based on the significance of the delta R^2 and then use an effect size cut-off. Including a qualifying sentence could help here. It could explain that this cut-off is valid for the window size used in the analyses (which it actually is, considering power aspects and correction for 756075 parameter tests).

Thank you for pointing this out! We have added your suggestion in the WCLR methods section:

Page 18: “As a rather conservative effect-size cut-off, $R_{CC}^2 < 0.25$ should be a valid multiple testing approach as well, given the number of regressions tested (756075).”

Reviewer #2 (Remarks to the Author):

I would like to commend the authors for doing an excellent job at implementing the changes all the reviewers suggested. I'm a big fan of the paper in its current state, job well done! The work done to create this paper is enormous, and I believe it lays a great groundwork for any other future papers that want to analyze mimicry in this way. The results weren't significant, but I think that's interesting in its own right - more mystery about how stress contagion occurs! - and I think your paper sets a good example of reporting important work in a high impact journal, even if those results aren't significant. In my view this paper is very

significant and I am looking forward to see this published.

Thank you!

Reviewer #3 (Remarks to the Author):

I am content with the author's responses to my concerns and appreciate the inclusion of some of my suggestions. I have no further concerns and would recommend a publication.

Thank you!